# Travel and terrain advice statements in public avalanche bulletins: A quantitative analysis of who uses this information, what makes it useful, and how it can be improved for users

Kathryn C. Fisher[1], Pascal Haegeli [1], Patrick Mair[2]

[1]School of Resource and Environmental Management, Simon Fraser University, Burnaby, V5A 1S6, Canada
[2]Dept. Psychology, Harvard University, Cambridge, 02138, United States

*Correspondence to*: Pascal Haegeli (pascal_haegeli@sfu.ca)

**Abstract.** Recreationists are responsible for developing their own risk management plans for travelling in avalanche terrain. To help recreationists mitigate their exposure to avalanche hazard, many avalanche warning services include explicit travel

and terrain advice (TTA) statements in their daily avalanche bulletins where forecasters offer guidance about what specific terrain to avoid and what to favour under the existing conditions. However, the use and effectiveness of this advice has never been tested to ensure it meets the needs of recreationists developing their risk management approach for backcountry winter travel.

We conducted an online survey in Canada and the United States to determine which user groups are paying attention to the

TTA in avalanche bulletins, what makes these statements useful, and if modifications to the phrasing of the statements would improve their usefulness for users. Our analysis reveals that the core audience of the TTA is users with introductory level avalanche awareness training who integrate slope-scale terrain considerations into their avalanche safety decisions. Using a series of proportional odds ordinal mixed effect models, we show that reducing the jargon used in the advice helped users with no or only introductory level avalanche awareness training understand the advice significantly better and adding an additional

explanation made the advice more useful for them. These results provide avalanche warning services with critical perspectives and recommendations for improving their TTA so that they can better support recreationists who are at earlier stages of developing their avalanche risk management approach and therefore need the support the most.

## 1 Introduction

Mountainous areas with untracked powder slopes are popular destinations for winter backcountry recreationists including

backcountry skiers and snowboarders, mountain snowmobile riders, and snowshoers. Even though detailed information on participation in winter backcountry recreation is sparse, there is strong anecdotal evidence that increasing numbers of people are taking to the mountains to pursue their mountain objectives, exercise, or simply enjoy nature (e.g., Birkeland et al., 2017; Techel et al., 2016; Winkler et al., 2016). However, recreating in the backcountry comes with serious risks. In North America alone, avalanches were responsible for the deaths of 334 recreationists between 2011 and 2020, and an unknown number of

injuries and near-misses (Avalanche Canada, 2019; CAIC, 2020). To safely recreate in avalanche terrain, recreationists must continuously monitor the severity of avalanche hazard and make informed decisions about what type of terrain is acceptable

to travel in under the current conditions (Canadian Avalanche Association, 2016). While some recreationists hire certified mountain guides to manage the risk from avalanches for them, most make their own decisions about when, where, and how to travel in the backcountry.

Having a good understanding of the existing avalanche conditions is critical for putting together a meaningful avalanche risk management approach for a trip into the backcountry. To assist recreationists with this process, most western countries with mountainous regions have public avalanche warning services that publish daily avalanche condition reports, commonly known as 'avalanche bulletins' or 'avalanche forecasts.' The main objective of these condition reports is to inform the reader about the severity of the existing avalanche hazard, which, in the context of public avalanche forecasting, is defined as the potential

for avalanches to cause harm to backcountry recreationists (Statham, 2008). In North America, public avalanche forecasters assess avalanche hazard according to the conceptual model of avalanche hazard (Canadian Avalanche Association, 2016; Statham et al., 2018a). Based on the available weather, snowpack, and avalanche observations, forecasters develop a picture of the types of existing avalanche problems, the locations where these problems can be found in the terrain, the likelihood of associated avalanches, and their expected destructive size (Statham et al., 2018a). This information is then summarized into a

set of three danger ratings that describe the overall severity of the conditions in the three elevation bands alpine, treeline and below treeline according to the North American public avalanche danger scale (Statham et al, 2010). Reflecting this process, avalanche bulletins in North America present the avalanche hazard information to their readers in a pyramid-like structure with the overall hazard rating given first, then details of avalanche problems, and finally additional details about snowpack structure, avalanche observations, and weather conditions. Avalanche bulletins in Europe use a similar but slightly different

structure (EAWS, 2021).

While avalanche bulletins provide an expert assessment of the existing hazard, recreationists must manage the associated risk by controlling their hazard exposure through their choices about when and where to go into the backcountry. These decisions can be made at different levels of sophistication, which were recently described in the bulletin user typology of St. Clair, Finn and Haegeli (2021). Bulletin User Type B, for example, exclusively base their decision to go into the backcountry at all on the

55 danger rating, whereas Type D use the avalanche problem information to distinguish between suitable and unsuitable areas for travel. A follow-up survey study by Finn (2020) showed that while bulletin users generally have a decent understanding of the concepts presented in the bulletin, roughly half of his survey participants exhibited challenges applying the information in a hypothetical slope evaluation task. This highlights that there might be a considerable gap between understanding the hazard information and combining it with terrain selection to make good risk management decisions.

There are several existing avenues through which recreationists can develop skills in forming a risk management plan and learn about selecting terrain to reduce exposure. Avalanche awareness courses taught by mountain guides and avalanche educators offer an important resource for recreationists to learn about practical avalanche risk management skills that can be used to understand both avalanche hazard and how to control risk through terrain selection. This was confirmed by Finn (2020), who found a strong association between the avalanche awareness training level of survey participants and their performance

at evaluating appropriate slopes for travel. To further assist recreationists in selecting appropriate terrain, various products

have been developed including specialized maps, decision aids, and web applications. For example, Statham et al. (2006) developed the avalanche terrain exposure scale (ATES) to describe the severity of backcountry trips with respect their general exposure to avalanche hazard using the three ordinal categories 'simple', 'challenging', and 'complex'. This expert terrain rating system has been used extensively to rate backcountry recreation areas in Canada (see https://www.avalanche.ca/planning/trip-planner), but ATES has also been applied in Norway (Larsen et al., 2020), Spain (Gavalda et al., 2013) and Switzerland (Pielmeier et al., 2013). While the ATES system provides an expert assessment of the terrain, Harvey et al. (2018) took a more physical process-oriented approach to classifying terrain when developing avalanche terrain maps based on GIS algorithms that explicitly identify potential avalanche release areas, possible runout zones, areas with the potential for remote triggering, and areas where small or medium-sized avalanches might lead to serious injures or deep burials due to terrain traps.

In addition to these terrain classifications, various decision frameworks have been developed to help recreationists combine the hazard information provided in avalanche bulletins with terrain characteristics of intended trips to make informed decisions about avalanche risk. Examples include the ground-breaking Reduction Method developed by Munter (1997), which combines the published danger rating with several terrain characteristics and group factors to determine whether the associated risk is acceptable, and the Avaluator Trip Planner (Haegeli, 2010), which combines the danger rating of the bulletin and the ATES rating of an intended trip graphically to provide users with guidance about what level of training and experience is required to effectively manage avalanche risk under the given conditions. Most recently, some of the concepts presented by these decision aids have been implemented as web applications. Avalanche Canada has an online trip planner that displays Avaluator assessments for selected recreation areas based on their ATES ratings and the current avalanche danger rating (https://www.avalanche.ca/planning/trip-planner), and the Swiss skitourenguru.ch website has implemented a version of the reduction method to provide detailed daily risk assessments of backcountry routes in the central European Alps (Schmudlach and Köhler, 2016).

The terrain classification systems and decision aids described above exist separate from the hazard information in avalanche bulletins, provide only generic guidance, and their application requires some training and experience. However, many avalanche bulletins also include travel and terrain advice (TTA) statements where avalanche forecasters directly communicate with their users to offer guidance about what specific terrain to avoid and what to favour under the existing hazard conditions. Avalanche warning services have taken a varied approach to including TTA statements in their bulletins. The Northwest Avalanche Center in Washington State, for example, presents the advice as part of their "bottom line" summary at the top of their bulletin webpage, while the Colorado Avalanche Information Center presents the information after the avalanche danger rating (NWAC, 2021; CAIC, 2021). In contrast, Swiss avalanche bulletins include the information along with the general hazard description (SLF, 2021). Avalanche Canada historically included the TTA statements on the avalanche problem tab but has moved them below to the danger rating at the beginning of the 2020/2021 winter season. These statements are the primary source of information on appropriate terrain selection found in avalanche bulletins.

Despite the important role of TTA statements for guiding users towards an appropriate risk management plan by linking daily hazard and terrain selection, there have been no studies to-date that specifically examine how these statements of advice are used by recreationists. In this study, we address this knowledge gap by:

> 1) Identifying which users pay the most attention to the TTA information
>
> 2) Examining factors that contribute to the usefulness of a TTA statement
>
> 3) Testing how simple modifications could increase the usefulness of these statements.

## 2 Methods

In the spring of 2020, we conducted a large-scale online survey to empirically examine different options for improving communication of hazard and terrain information in avalanche bulletins. This paper focuses on the results pertaining to the travel and terrain advice (TTA) statements, whereas additional analyses investigating information graphics and bulletin interactivity are presented in Fisher et al. (2021) and Fisher et al. (2022) respectively.

### 2.1 Survey Design

To investigate the primary audience of the TTA statements in the bulletin, we asked all survey participants how much attention they generally pay to the TTA statements. This was to better understand which users are engaging with the TTA, as well as to target subsequent questions about the TTA towards participants who actually use it. Users were asked to rate their attention to the TTA on a four-level ordinal scale of 'None', 'A little', 'A considerable amount', and 'A large amount'. Users who selected any response other than 'None' were directed towards a section with more detailed questions about specific TTA statements.

We created a database of 18 TTA statements (Appendix A) drawn from a larger database of statements provided by Avalanche Canada. The 18 statements selected covered a variety of snow conditions, terrain features, or behaviors participants should be mindful of while recreating in avalanche terrain. We also ensured the statements represented a mix of communication styles (see 'statement type' column in Table A1) including direct recommendations for actions, mindsets to adopt while traveling, or simply bringing attention to certain key features. For each statement, the research team created a second statement that altered the original statement to vary the amount of jargon in the statement or add additional explanatory details about condition described in the statement. Additional details included more detailed descriptions of the impacts of a condition or information on how to identify a feature into the statement. The end result was a database of 36 statements divided across four treatments: 'more jargon', 'less jargon', 'no explanation', and 'added explanation'. This structure allowed us to compare the impact of the statement treatment while controlling for the subject of the statement.

Each participant was shown three TTA statements drawn semi-randomly from the database of 18 paired statements. Each participant saw a combination of original and modified statements, and the survey structure was designed so that individual participants were not presented with both the original and modified versions of the same statement.

To comprehensively capture participants' perspective of the statements, we asked participants to rate each of the presented statements with respect to three different aspects (Figure 1). First, if the TTA included a key phrase (e.g., 'minimize exposure', 'hard wind slab', 'thick melt-freeze surface crust'), the phrase was highlight and participants were asked how easy it was to understand the phrases on a six-level scale including 'Very difficult', 'Difficult', 'Somewhat difficult', 'Somewhat easy', 'Easy', and 'Very easy'. All but two TTA statements included this question. Second, if the key phrase described a snow condition or terrain feature that users need to recognize in the field to apply the statement meaningfully, participants were asked how confident they were about recognizing the highlighted condition in the field on a five-level scale with response options including: 'not at all confident', 'somewhat confident', 'fairly confident', 'very confident', and 'extremely confident'. This question was only included with six pairs of TTA statements. Finally, for all statements, participants were asked how useful they thought the statement was overall for their avalanche risk management practices using a five-level scale including 'Not at all useful', 'Somewhat useful', 'Fairly useful', 'Very useful', and 'Extremely useful'. The aim of this three-question setup was to provide deeper insight on why TTA statements are considered useful (or not) and how that perspective is affected by our statement alterations.

> **TRAVEL ADVICE STATEMENT 2**
>
> Watch for areas of **hard wind slab** on alpine features. A good indicator is when travel suddenly gets easier because you do not sink in as much.

- **How easy or difficult do you find the highlighted condition description to understand?**
  *Please select one of the following options.*

  ○ Very difficult
  ○ Difficult
  ○ Somewhat difficult
  ○ Somewhat easy
  ○ Easy
  ○ Very easy

- **How confident are you in your ability to recognize this condition in the field?**
  *Please select one of the following options.*

  ○ Not at all confident
  ○ Somewhat confident
  ○ Fairly confident
  ○ Very confident
  ○ Extremely confident

- **Overall, how useful do you find this travel advice statement for your decision-making in avalanche terrain?**
  *Please select one of the following options.*

  ○ Not at all useful
  ○ Somewhat useful
  ○ Fairly useful
  ○ Very useful
  ○ Extremely useful

**Figure 1: Screen shot of survey question for example statement**

The survey contained additional background questions so that we could contextualize and identify patterns among respondents. We drew from questions included in Finn's (2020) survey and incorporated questions about participants' primary modes of winter recreation in the backcountry, how many years and days per year of experience they had, and their bulletin user type as described by St. Clair (2019). Further questions collected basic sociodemographic items including self-identified gender, age, education level, location of residence. Additional sections included in the survey to address the other research questions are described in Fisher et al. (2021) and Fisher et al. (2022).

The survey was developed during the early part of the 2019/20 winter season and extensively tested in February and March 2020 prior to release. Survey testing began with an initial round of testers with moderate to high levels of winter backcountry recreation experience and avalanche industry experts. A second round of testing included users from novice to expert participants. The survey was also reviewed and approved by the Office for Research Ethics of Simon Fraser University (SFU ethics approval 2020s0074).

## 2.2 Recruitment and Survey Development

The primary target audience for our survey was North American avalanche bulletin users, which we recruited in a variety of ways. The foundation of our recruitment were 3,047 bulletin users who participated in previous avalanche bulletin surveys conducted by our research group and indicated that they were interested in participating in future studies. The survey was officially launched on March 23, 2020, by sending invitation emails to 300 individuals from this existing panel of prospective participants. This soft launch allowed us to monitor the initial responses and address any survey issues if necessary. However, the survey worked as designed and no modifications were required. On March 26, 2020, we sent invitation emails to the rest of our panel of prospective participants (2,747 individuals) and between March 26 and April 1, 2020, the survey was also actively promoted by our partnering avalanche warning services (Avalanche Canada, Parks Canada, Colorado Avalanche Information Centre, Northwest Avalanche Center). Each of these warning services helped us recruit participants by including a banner on their bulletin website and promoting the survey through their social media channels. We also advertised our study by posting on various social media sites popular among winter backcountry users, such as *South Coast Touring* and *Backcountry YYC* on Facebook, and by reaching out to community leaders to distribute the survey among their followers.

The survey sample for the present analysis was drawn on May 31, 2020, after which no additional surveys were included in analysis. At the close of the survey, 6789 individuals had visited our survey and 3,668 (55.3%) completed it. The vast majority of the dropouts (1,829, 27.6%) did not continue after looking at the first page of the survey that described the objective of the study and structure of the survey. The dropout rate for individual survey pages was 1% or less except the page that introduced the route-ranking task (57, 3.4%). Of the individuals who completed the survey, 1,600 (44.6%) were participants of previous survey studies of our research group who received an invitation email. Other substantial recruitment sources included announcements on avalanche bulletin websites (17.5% of participants who completed survey), social media posts by collaborating avalanche warning services (9.2%), and other posts in social media groups (e.g., Facebook, Instagram) focused on winter backcountry recreation (21.5%).

## 2.3 Data Analysis

Our analysis approach started with the use of standard descriptive statistics to describe the nature of the analysis dataset and explore the relationships between different variables. We used a standard proportional odds ordinal regression model to evaluate how much attention users paid to the TTA in general, but since each of our participants evaluated multiple statements, we employed a series of proportional odds ordinal mixed effects regression models to explore how participants rated their understanding of key phrases highlighted in the statements, how confident they felt recognizing those conditions in the field, and how useful they found the statements overall. Mixed effects models are a type of regression model that accounts for correlations that emerge from repeated measure designs or nested data structures (Harrison et al., 2018; Zuur et al., 2009). To accommodate these data structures, mixed effect models include both fixed and random effects in the regression equations. The fixed effects, which are equivalent to the intercept and slope estimates in traditional regression models, capture the

relationship between the predictor and response variables for the entire dataset. While traditional regression models assign the remaining unexplained variance in the data (i.e., randomness) entirely to the global error term, mixed-effect models partition the unexplained variance that originates from groupings within the dataset into random effects. Thus, random effects highlight

how groups within the dataset deviate from the overall pattern described by the fixed effects included in the model. While there is some judgment involved in deciding what predictors are included in the model as a fixed or random effect, it is generally the grouping variables that are not explicitly of interest that enter the analysis as random effects. In our analysis, this includes the participants as they assessed three TTA statements each, as well as the 18 pairs of original and modified versions of the TTA statements.

Since the TTA statements included in our study had their wording modified to either reduce jargon (11 statements; Table A1) or include additional explanations (7 statements; Table A1) but not both, we estimated two sets of three ordinal regression models to separately examine the effect of these modifications on the understandability of the statements, participants' confidence to recognize the conditions in the field, and the overall usefulness of the statements. All analyses required the use of mixed effect models except the model examining participants' confidence recognizing the condition when an additional

explanation was added because we only had a single TTA statement where this question was relevant (see Table A1: Statement 8). For this analysis, we used a standard ordinal logistic regression model instead.

To explore our main research question, we included the predictor variables of 'statement treatment' (less jargon, more jargon, no explanation, added explanation) and 'avalanche training' (none, introductory, advanced, professional) as fixed effects in all of our regression models by default. Since we were interested in better understanding how the different statement treatments

affect the responses of participants with different levels of training, we also included this interaction in the models for all three questions. In addition, we included 'statement type', 'years of experience', 'days per winter in backcountry', 'bulletin user type', and 'country of residence' in the initial models by default but removed them if their parameter estimates did not reveal a significant influence on the model (i.e., p-values > 0.050). However, we also took the magnitude of the observed effect into account for deciding whether including the parameter was meaningful. After an initial model was estimated, the parameter

estimates of ordinal predictor variables were examined for linear relationships, and if a linear relationship was present and meaningful, the specific ordinal predictor variable was replaced with a numeric variable to produce a more parsimonious model. Because our experimental setup included more statements where we modified the amount jargon, we estimated the jargon models first, and subsequently estimated the models for examining the effect of the added explanation with the same parameters settings. This ensured that both analysis streams considered the same covariates and were comparable despite

varying sample sizes.

The distributions of many of our predictor and response variables were considerably skewed with many participants selecting options higher on the scale (e.g., Use of TTA: 'A considerable amount' or 'A large amount') and only few choosing lower options (e.g., Use of TTA: 'Not at all'). To address this issue, we examined the distributions of all variables prior to estimating the models and combined categories with extremely small counts (e.g., < 1%). This resulted in the following changes in our

variables: a) we combined '1-2 days' with '3-10 days' into a single category in our backcountry days per winter predictor

variable, and b) we merged 'Very difficult' and 'Difficult' into a single category for the level of understanding rating. While these changes help to even out the distributions of the values in our variables, many of our response variables are still highly skewed. While these distributions contain meaningful insight, it can be difficult for ordinal regression models to properly represent them, and it is possible that categories with very low response frequencies are overpredicted. To examine whether our models describe the observed responses adequately, we performed a posterior predictive check (Gelman and Hill, 2007, p. 158) where we simulated the distributions of the response variable using the fitted model and then compared them to the observed data. Our results indicated that all of our models capture the observed frequencies of the response variables nicely, and there was no need to further combine response categories.

Since our models include predictor variables with considerable correlations (e.g., avalanche awareness training and bulletin user type), there is the risk of variance inflation, which affects the standard errors and produces incorrect p-values. To explore the impact of correlations on the standard errors of the regression parameters, we computed the generalized variance inflation factor (GVIF) and used the general rule of thumb for categorical variables that $GVIF^{1/(2 \cdot DF)} < 5$ to assess whether there is any issue (Fox and Monette, 1992).

We conducted our entire analysis in R (Version 4.1.3; R Core Team, 2022) and used the `clmm` function of the `ordinal` package (Christensen, 2019) to estimate our ordinal mixed effects models and the `polr` function of the `MASS` package (Venebles and Ripley 2002) to estimate our standard ordinal logistic regression models. While we performed the posterior predictive check manually since the `posterior_predictive_check` function in the `performance` package (Lüdecke et al., 2021) has not been implemented for our model types, we used the `gvif` function from the `car` package (Fox and Weisberg, 2019) to calculate the generalized variance inflation factors. Since neither of these checks can be performed on mixed effects models, we also estimated standard ordinal regression models with the same parameter setting for all mixed effects models and used those to simulate the distribution of the response variable and calculate the GVIF.

Since parameter estimates of ordinal logistic regression models are notoriously difficult to interpret directly, we used effects plots that show the probabilities for selecting specific levels of the response variable to illustrate the results. We used the `ref_grid` and `emmeans` functions of the `emmeans` package (Lenth, 2019) to both estimate these probabilities and conduct post-hoc pairwise comparisons to explicitly test for significant differences between different combinations of predictor variables. To counteract the issue of Type I error inflation from multiple comparisons, p-values were calculated with the default correction approach implemented in the function. When reading about the results and examining the effects plots, it is important to remember that the shown probabilities are calculated for a specific combination of predictor values and cut point in the response variable to illustrate a particular pattern. Hence it is more important to look at the general pattern and significance of the differences in these probabilities than their absolute values as they change depending on the chosen predictor values.

## 3 Results

### 3.1 Participant Demographics

To ensure meaningful results, we only included participants in our analysis dataset who completed all pages of the survey, whose reported residence was in Canada or the United States, who were over the age of 20,[1] and whose choices for primary activity and avalanche awareness training aligned with the predefined options. In addition, we excluded participants who took less than 10 minutes or more than 2.5 hours to complete the survey, and participants who did not respond to the question about how much attention they pay to the travel and terrain advice (TTA). We also disqualified participants who spent less than 30 seconds or more than 10 minutes viewing the travel advice page to remove participants who just clicked through it or got interrupted while completing the page. We also eliminated participants who did not provide information on their years of backcountry experience and how many days they spend in the backcountry each year as they play a critical role in our analysis. Finally, we removed participant whose self-reported bulletin user type was A or F due to the low number of participants in each group, as well as to reduce correlation among variables. The final analysis dataset consisted of 2,998 participants, which represented 81.7% of the 3,668 individuals who completed the survey. These participants provided a total of 8,900 TTA statement assessments. However, the datasets for the individual analyses vary as not all three assessment questions were relevant for every TTA statement.

Of the 2,998 participants, 76.6% identified as male (2,273 participants), 36.8% were between 25 and 34 years old (1,101 participants), 80.4% had a university-or-higher education (2,304 participants) and 82.2% had completed at least an introductory avalanche safety training course (2,465 participants). Backcountry skiers represented the highest proportion of recreationists in the study with 78.1% of the sample (2,341 participants) identifying backcountry skiing as their primary backcountry winter activity. Additional types of recreationists present in our sample included out-of-bounds skiers (7.7%, 230 participants), snowshoers (5.8%, 173 participants), and snowmobilers (5.0%, 151 participants), and less than two percent ice climbers and snowmobile-accessed backcountry skiers. The largest group of participants (31.6%, 947 participants) were relatively new to their sport, with between 2 and 5 years of experience. However, the distribution of years of experience was relatively even with 19.7%, 19.1% and 25.0% of the sample stating that they had 6-10 years, 11-20 years, and more than 20 years of backcountry experience respectively. Only 4.5% of the sample (136 participants) reported that this was their first year of backcountry recreation. Bulletin user types 'D - Distinguish and Integrate Avalanche Problem Conditions' and 'E - Extends Analysis' made-up 30.3% and 47.8% of participants respectively (909 and 1434). While we observed a significant correlation between avalanche training and bulletin user type (Spearman rank correlation: 0.350; p-value < 0.0001), the analysis sample included a range of training levels at each bulletin user type (Table 1). Finally, 70.4% (2,110) of responses were from residents of the United States.

---

[1] Participants younger than 20 were excluded from the analysis since that age category could include minors, and the survey did not allow us to get consent from a parent or legal guardian.

**Table 1: Distribution of avalanche training levels with respect to self-identified bulletin user type within the final dataset. Percentage values are row percentages except in the total column where they represent column percentages.**

| Bulletin user type | No training | | Introductory level | | Advanced level | | Professional level | | Total | |
|---|---|---|---|---|---|---|---|---|---|---|
| Type B | 87 | (53%) | 67 | (41%) | 8 | (5%) | 3 | (2%) | 165 | (6%) |
| Type C | 171 | (35%) | 241 | (49%) | 54 | (11%) | 24 | (5%) | 490 | (16%) |
| Type D | 122 | (13%) | 531 | (58%) | 176 | (19%) | 80 | (9%) | 909 | (30%) |
| Type E | 153 | (11%) | 591 | (41%) | 351 | (25%) | 339 | (24%) | 1434 | (48%) |
| Total | 533 | 18% | 1430 | 48% | 589 | 20% | 446 | 15% | 2998 | (100%) |

## 3.2 Attention to Travel and Terrain Advice

Of the 2,998 participants included in the analysis dataset, 52.3% (1,569) stated that they pay a large amount of attention to the TTA statements in the avalanche bulletin (scale: 'none', 'a little', 'a considerable amount', and 'a large amount'). Thirty-nine percent (1,169) respondents stated that they pay a considerable amount of attention to the TTA, 8.1% (244) indicated that they

only pay a little bit of attention to the TTA, and less than 1% (16) responded that they pay no attention to the TTA.

Our ordinal regression model for the probability of participants' response selections revealed four significant predictors, which included the bulletin user type of the participant, the level of avalanche training they had completed, how many days they spend per year engaged in their preferred backcountry activity, and their country of residence (Table B1). Initial model explorations showed that the effect of number of days in the backcountry per winter was strongly linear, and we decided to

295 replace this ordinal variable with a numerical one to produce a more parsimonious model. The posterior predictive check showed that the skewed distribution of the response variable was well captured by our model, and all of the generalized variance inflation factors were well below 5 indicating that there were no issues with predictor correlations.

Participants who self-identified as bulletin user Type D were the most likely to pay attention to the TTA statements. Figure 2a illustrates this effect by showing the estimated marginal probabilities for selecting 'A large amount' for the different bulletin

user types with avalanche awareness training set to introductory, 11-20 days in the backcountry per winter, and Canada as the country of residence. Using these parameter values, the model estimates a 57.0% chance that Type D users response that they pay 'A large amount' of attention to the advice, followed by Type E users at 52.1%. This difference was not statistically significant (post-hoc pairwise comparison: p-value = 0.0663) even though the p-value is close to the 5% threshold. However, Type C and B users were significantly less likely to indicate that they pay 'A large amount' of attention to the TTA than

Type D users (42.4% and 41.6% respectively, both p-value < 0.0001).

In addition to the bulletin user type, the level of avalanche training a participant had completed was also a significant predictor of how much attention they pay to the TTA statements (Figure 2b). Participants with professional level training were significantly less likely to report that they pay 'A large amount' of attention to the TTA statements (41.0%) than participants

with advanced level training (52.0%, p-value = 0.0007), which was no different than participants with introductory training (57.0%, p-value = 0.1197) or no training (54.1%, p-value = 0.5914).

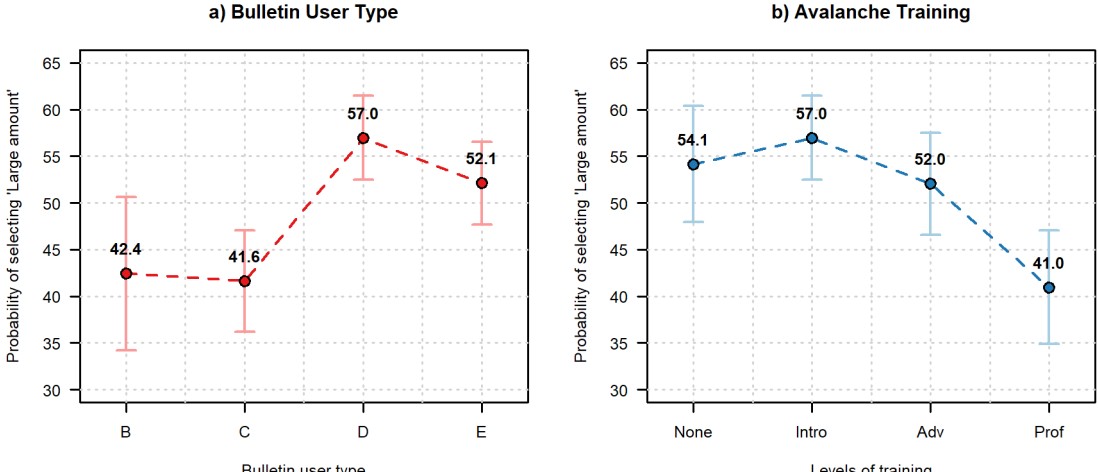

**Figure 2: Estimated marginal probabilities for selecting 'A large amount' for a) different bulletin user types, and b) different avalanche training levels. Error bars represent 95% confidence intervals for probabilities calculated from the subsample for the particular parameter level. Variable levels for estimated marginal probabilities calculations: bulletin user type: Type D (right panel only); avalanche awareness training: introductory (left panel only); days in backcountry per winter: 11-20 days; country of residence: Canada.**

Another predictor of participants' attention to the TTA included the average number of days they spend in the backcountry during a typical winter, which we interpreted as their level of engagement in the activity. Participants who spend more days in the backcountry were more likely to indicate lower levels of attention to the TTA, while participants who spend fewer days in the backcountry are likely to state that they pay more attention to the TTA. Finally, participants residing in the United States were more likely to indicate higher levels of attention to the TTA than Canadian residents.

## 3.3 Overall Usefulness of Travel and Terrain Advice

Our dataset for this analysis consisted of 8,900 usefulness ratings. Most participants found the TTA useful, with 54.8% of participants reporting that they found the statements either 'Very useful' or 'Extremely useful' (scale: 'Not at all useful', 'Somewhat useful', 'Fairly useful', 'Very useful', and 'Extremely useful'), and only 2.9% (262) of the ratings were 'Not at all useful'. Of the available usefulness assessments, 5,529 (62.1%) related to statements where the amount of jargon was modified, and 3,371 (37.9%) assessments are linked to statements where explanations were added. Overall, the statements in the jargon

sample were rated significantly more useful than the explanation statements (58.3% versus 49.1% rating them 'Very useful' or 'Extremely useful'; Kruskal-Wallis rank sum test: p-value < 0.001).

We built separate ordinal mixed effect regression models to understand the influence of jargon and added explanation on the reported usefulness with participant ID, statement ID and statement version code as random effects. Prior to estimating the models, we created explicit 'Not applicable' categories for the level of understanding and confidence in recognition variables since these ratings were not relevant (and therefore not asked) for all TTA statements included in the study (Table A1). This allowed us to include the entire dataset in the analysis. Overall, 1,081 (12.1%) TTA statement assessments did not have a level of understanding rating, and 5,564 (62.5%) statements did not have a confidence in recognition rating. Note that all the 'Not applicable' values for the level of understanding ended up in the explanation model and none of them in the jargon model. Hence, there is no parameter estimate for 'Not applicable' in the jargon model.

Our final model for examining the influence of jargon included five predictors (statement treatment, level of avalanche training, attention to TTA statements, level of understanding, and recognition confidence) as main effects and the interaction effect between statement type and level of avalanche awareness training. We used the same predictor variables for examining the effect of the added explanation to make sure that the results are comparable. Readers interested in more details are invited to examine the precise model specifications in the accompanying R code published in Haegeli et al. (2022). Our posterior predictive checks indicated that the skewed distributions of the response variable were well captured by our models, and the low generalized variance inflation factors confirmed no issues with predictor correlations.

Overall, the two models examining what makes a TTA statement useful revealed very similar main effect patterns with the results of the jargon model generally being more significant due to the larger sample size (Table B2). Hence, our presentation of the results focuses on the jargon model unless explicitly stated. The strongest predictor that emerged from our analysis was how well participants understand the statement. Figure 3a illustrates this effect by presenting estimated marginal probabilities for selecting 'Very useful' or 'Extremely useful' as a function of the level of understanding calculated for participants with introductory level avalanche awareness training, 6-10 years of winter backcountry experience, who spend an average of 11-20 days in the backcountry each winter, who pay considerable attention to the TTA statements, and are fairly confident in their ability to recognize the condition in the field. In this scenario, participants who found a statement 'difficult' to understand had the lowest percent chance of finding the statement 'Very useful' or 'Extremely' useful (1.7%). However, with every increase in rating of how easy it is to understand the statements the chance of a high usefulness rating becomes significantly higher. Participants who rated the advice as 'somewhat difficult' have a 7.5% chance of finding the advice 'Very useful' or 'Extremely useful', but the percentage value jumps to 23.5% for participants who found it 'Somewhat easy', and up to 51.1% (p-value < 0.0001) and 80.5% (all consecutive post-hoc comparisons: p-value < 0.0001) for participants who found it 'Easy' or 'Very easy' to understand the statements.

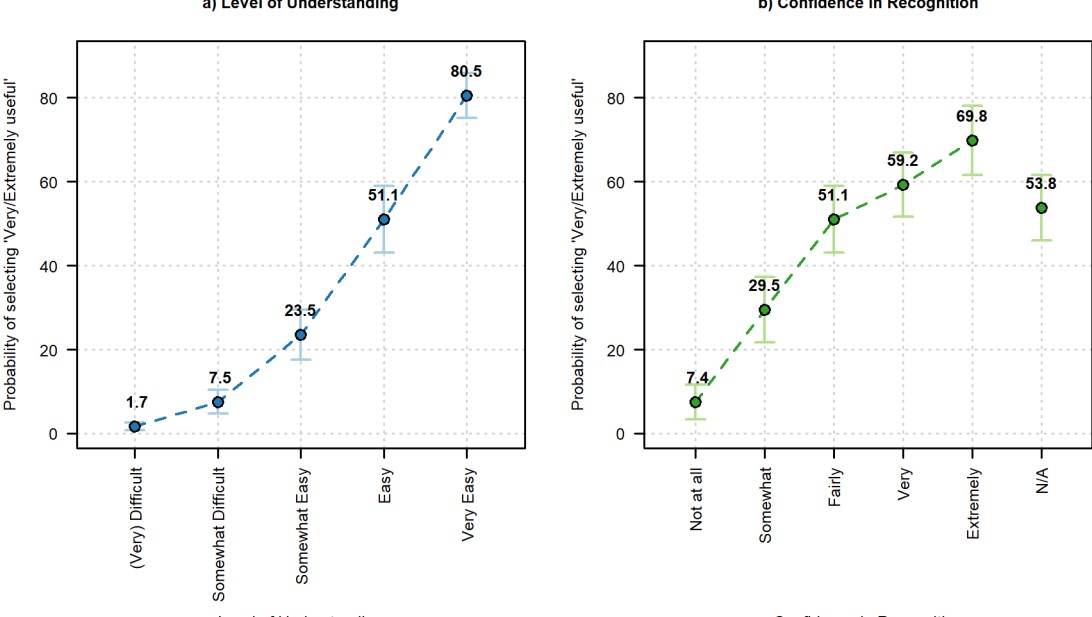

**Figure 3. Estimated marginal probabilities for selecting 'Very useful' or 'Extremely useful' for usability of statement as function of a) participants' level of understanding, and b) their confidence in recognizing the condition or feature. Error bars represent 95% confidence intervals for probabilities calculated from the subsample for the particular parameter level. Variable levels for estimated marginal probabilities calculations: avalanche awareness level: introductory; attention to TTA statements: considerable; years of backcountry experience: 6-10 years, average number of days in the backcountry per winter: 11-20 days; level of understanding: easy (right panel only); confidence in recognition: fairly confidence (left panel only).**

In addition to the ease of understanding, participants' confidence in their ability to recognize the condition in the field was also a significant predictor for how useful they find a statement (Figure 3b). Our estimated marginal means with the same settings as for the previous calculations for participants who find the statement easy to understand show that participants who were 'Not at all' confident in their ability to recognize a specific condition in the field only had an 7.4% chance of finding the statement 'Very useful' or 'Extremely useful', while participants who were 'Somewhat' confident had a 29.5% chance of the same responses (post-hoc pairwise comparison: p-value < 0.0001). This effect continues for higher confidence levels, with the percentage chance of finding the statements 'Very useful' or 'Extremely useful' rising to 51.1% (p-value < 0.0001) for participants expressing that they were 'Fairly' confident at recognizing a condition in the field, 59.2% (p-value = 0.0073) for those 'Very' confident, and 69.8% (p-value = 0.0036) for those 'Extremely' confident. The presented effect of the level of understanding and the recognition confidence on the perceived usefulness of the TTA statement were very similar and highly significant in both the jargon and explanation models. The factors influencing participants' level of understanding of the TTA statement and their confidence in recognizing the stated condition in the field are discussed in detail in Section 3.4 and 3.5 respectively.

Another factor that led to higher reported usefulness of the statements in both models was the amount of attention participants

pay to the travel advice in general (Table B2). As the attention increases, the percent chance a statement will be considered useful also increases.

In contrast to these similarities, how participants' usefulness ratings are influenced by participants' avalanche awareness education and the TTA statement treatment differs substantially between the two models. In the jargon model, avalanche awareness training emerged as a significant main effect with participants with professional level training finding the TTA

statements significantly less useful (Table B2 and Figure 4a). Somewhat surprisingly, there was no significant direct impact of the level of jargon on the participants' usefulness ratings, which is illustrated by the almost perfectly overlapping lines in Figure 4a. In the explanation model, on the other hand, avalanche awareness training did not have a direct effect on participants' usefulness ratings, but we found an interesting main and interaction effect for the added explanation (Figure 4b). Our estimated marginal probability calculations for participants who pay considerable attention to the TTA statements, find

them easy to understand and have fairly high confidence in their ability to recognize the conditions in the field show that the added explanation significantly increases the change of participants with introductory avalanche awareness training to rate the statement 'very or extremely useful' (46.7% versus 58.6%; post-hoc comparison: p-value $< 0.0001$). While the observed increase for participants with no and advanced avalanche awareness training was also substantial (approx. 7 percentage point increase), they were statistically only marginally significant with their p-values being close to the 5% threshold (0.0720 and

0.0613 respectively). The added explanation did not make a difference in the usefulness ratings of participants with professional level avalanche awareness training (p-value $= 0.8508$).

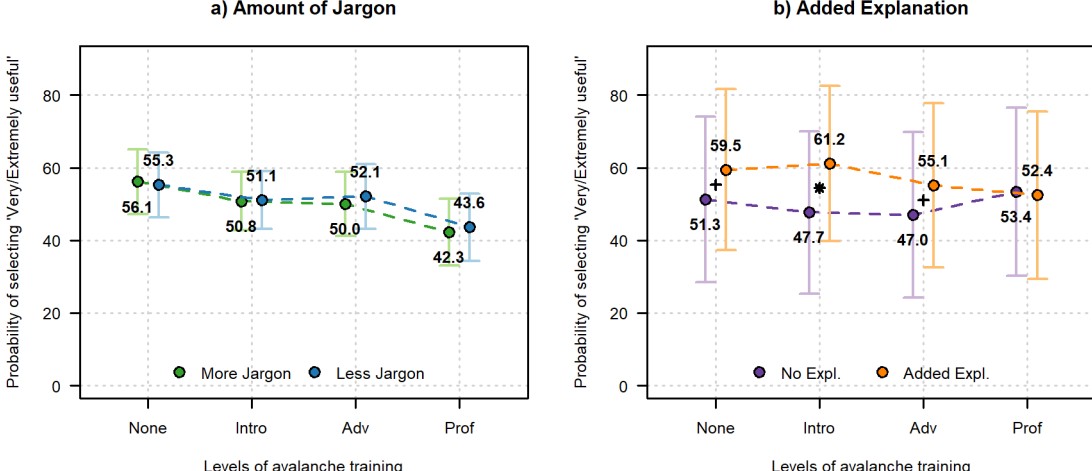

**Figure 4 Estimated marginal probabilities for selecting 'Very useful' or 'Extremely useful' for usability of statement as function of a) the interaction effect of avalanche training and amount of jargon, and b) the interaction effect of avalanche training and added explanation. Error bars represent 95% confidence intervals for probabilities calculated from the subsample for the particular parameter level. Significant post-hoc pairwise comparisons are indicated with asterisks (p-values < 0.01), diagonal crosses (0.01 ≤ p-values < 0.05) or horizontal crosses (0.05 ≤ p-values < 0.1). Variable levels for estimated marginal probabilities calculations: attention to TTA statements: considerable; level of understanding: easy; confidence in recognition: fairly confidence.**

In both models, there was greater unexplained variance associated with individual participants than with the statements used. This indicates that which specific statements participant saw did have a smaller impact on their responses compared to the variations in the nature of participations not accounted for in the model. This gives us confidence that the specific selection of statements used did not unduly impact our results. Additionally, statement type, which describes the nature of the TTA statement ('action', 'attitude', and 'fact'; see Appendix A) did not emerge as a significant predictor for the statements' usefulness ratings. This means that participants did not find any of the three types fundamentally more or less useful. Hence, the statement type parameter was removed from the model during the development of the model.

## 3.4 Understanding of Key Phrase

Participants provided a total of 7,819 understanding ratings, and overall, they found the key phrases highlighted within the travel and terrain statements easy to understand, with 70.5% of the ratings at 'Easy' to 'Very easy' to understand (scale: 'Very difficult', 'Difficult', 'Somewhat difficult', 'Somewhat easy', 'Easy', and 'Very easy'). As explained in Section 2.3, the levels of 'Very difficult' and 'Difficult' were combined due to a very low number of times 'Very difficult' was selected. Overall, only 3.4% of the rating selected the lowest two levels.

We built two ordinal mixed effects regression models with participant ID, statement ID, and statement version ID as random effects to explore how participants understand key phrases in TTA statements; one model examining on how jargon affects

understanding, and the other one studying the effect of added explanations. The sample size for the jargon model was 5529 ratings (70.7% of dataset), and the explanation dataset consisted of 2290 ratings (29.3%). Similar to the dataset used for the usability analysis presented in Section 3.3, the understandability ratings differed significantly between the two datasets. In the jargon dataset, 72.5% of the ratings were 'easy' or 'very easy' whereas the same ratings were only chosen 65.6% in the explanation dataset (Kruskal-Wallis rank sum test: p-value < 0.001).

Our final model for the analysis of jargon included six predictors (statement treatment, bulletin user type, level of avalanche training, reported level of attention to avalanche bulletin, years of experience in the winter backcountry, and the number of days spent in the backcountry each winter) as main effects and the interaction between the level of training and the statement treatment. (Table B3). We used the same predictor variables for examining the effect of the added explanation to make sure that the results are comparable. Readers interested in more details are invited to examine the precise model specifications in the accompanying R code published in Haegeli et al. (2022). Just like for the usefulness models, our posterior predictive checks and generalized variance inflation factors did not raise any concerns with the models.

Similar to the results presented in the previous section, the two models examining the level of understanding of key phrases revealed similar patterns with the results of the jargon model generally being more significant due to the larger sample size (Table B3). The significant predictors that emerged from the analysis are a combination of participant and statement characteristics. Significant participant characteristics included bulletin user type, avalanche awareness training (jargon model only), number of years of experience, average number of days spend in the backcountry each winter, and how much attention they pay to the TTA statements. The overall strongest predictor was participants' self-identified bulletin user type, and participants with more advanced bulletin use practices tended to find the key phrases easier to understand. In the jargon model, for example, the estimated marginal probabilities calculated for participants with introductory avalanche awareness training, 6-10 years of backcountry experience, who spend 11-20 days in the backcountry each winter and pay considerable attention to the TTA statements, nicely illustrate the trend. With these variable settings, the calculated chance of rating the understandability of the key phrase in TTA statements with less jargon as either 'Easy' or 'Very easy' was 61.7% of Type B users, 69.2.1% for Type C users, 76.7% for Type D users and 81.6% for Type E users. While the difference between Type B and C user was not statistically significant (post-hoc pairwise comparison: p-value = 0.1682), the subsequent consecutive pairwise comparisons were (C vs. D: p-value = 0.0028; D vs. E: p-value = 0.0028). This general increase was also detected in the explanation model, but there were stronger similarities between Types B and C and Types D and E, and the only significant consecutive pairwise comparison was between bulletin user Types C and D (p-value = 0.0027).

Similarly, the chance of rating the understandability of the key phrase 'Easy' or 'Very easy' generally increased with more years of experience and more days in the backcountry per winter. An interesting pattern emerged in the relationship between years of experience and the understandability rating in the jargon model (Figure 5a). Using the same settings for estimating marginal probabilities as above with the bulletin user type set to Type D, our analysis revealed that participants in their first year in the backcountry were significantly less likely to find the statements at least easy to understand than participants with 2-5 years of experience (65.9% v. 75.8%, p-value = 0.0416). The other cohorts for backcountry experience responded similarly

to the 2-5 years group, and there were no significant differences between them. The number of days participants spent in the backcountry each winter was also included as a predictor, and the likelihood participant found the phrase at least easy to understand increased significantly with more time spent in the backcountry.

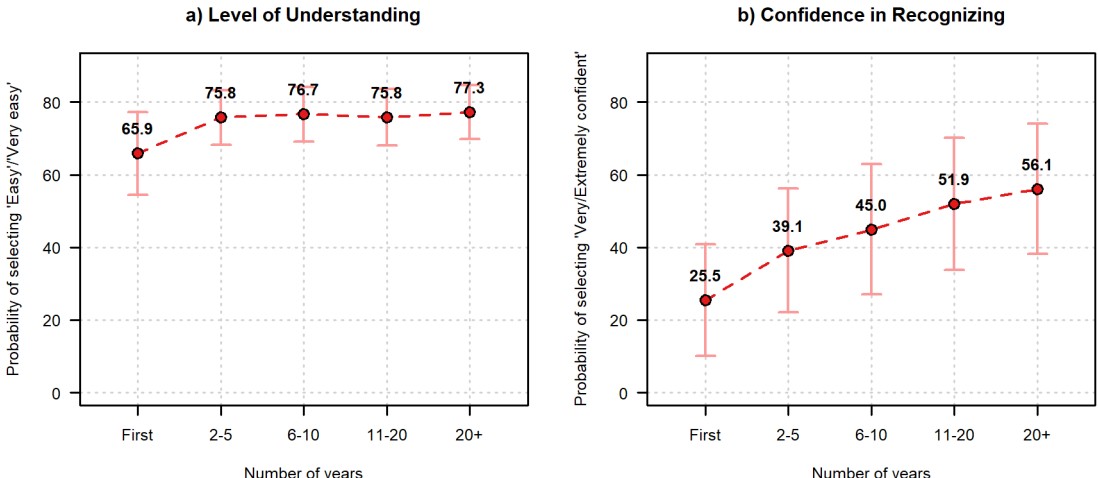

**Figure 5: Estimated marginal probabilities for selecting 'Easy' or 'Very Easy' for understandability of statement (left panel) and selecting 'Very' or 'Extremely' for confidence in recognizing the key phrase in the field as function of years of backcountry experience in the jargon model. Error bars represent 95% confidence intervals for probabilities calculated from the subsample for the particular parameter level. Variable levels for estimated marginal probabilities calculations: attention to TTA statements: considerable; days spend in the backcountry each winter: 11-20 days; avalanche awareness training: introductory; bulletin user**
**type: Type D; statement treatment: less jargon.**

Finally, how much attention participants generally pay to the TTA statements also had a positive effect on participants understandability ratings. Participants who pay higher amounts of attention to the TTA tended to find the statements easier to understand. While correlations naturally exist between some of these participant characteristics (e.g., a participation with more
475 years of experience might also use the bulletin in a more sophisticated way), our examination of the generalized variance inflation factors indicated that there are no issues in our parameter estimations.

The level of jargon had a significant main effect on how participants rated their understanding of the highlighted phrases shown in in the TTA statements (p-value: 0.0224; Table B2). However, this effect was modulated by the interaction effect with the level of avalanche training a participant had completed (Figure 6a). Professionals and recreationists with advanced level
training were overall the most likely to say they find the key phrases 'Easy' or 'Very easy' to understand, and the estimated marginal probabilities[2] did not differ significantly between the version with less jargon or the version with more (Advanced: 75.5% vs. 74.5% p-value = 0.8070, Professional: 78.8% vs. 78.5%, p-value = 0.9402) However, among participants with no

---

[2] Estimated marginal probabilities for the ease of understanding model were calculated using the following parameter levels: bulletin user type: Type D; avalanche awareness training: introductory; year of experience: 6-10 years; average days in backcountry per winter: 11-20 days; and attention to travel advice: considerable.

training or introductory recreational training, it was significantly more likely that they would find the statements easy to understand if presented with a version that had less jargon (No Training: 76.8% vs. 21.5%, p-value = 0.0246, Introductory Training: 76.7% vs. 67.4%, p-value = 0.0166). Crucially, the post-hoc pairwise comparisons show that with the statement versions with less jargon do not show any significant difference in the ease of understanding ratings across the different training levels. In other words, all training levels reported the same ease of understanding for the less jargon-filled statements.

The explanation model revealed that the added explanation had a similar effect on participants' level of understanding rating as the reduced jargon (Figure 6b). The combination of the significant main effects for avalanche awareness training and statement treatment together with their marginally significant interaction effect (Table B3) produced a response pattern where the understandability of statements without the additional explanation increased with avalanche awareness training, and the benefit of the added explanation was biggest for participants without any training (Figure 5b: 70.4% vs. 57.0%; p-value = 0.0090). Note that the errors bars for the explanation model are much larger because of the smaller sample size, but they are not directly indicative for the post-hoc comparisons.

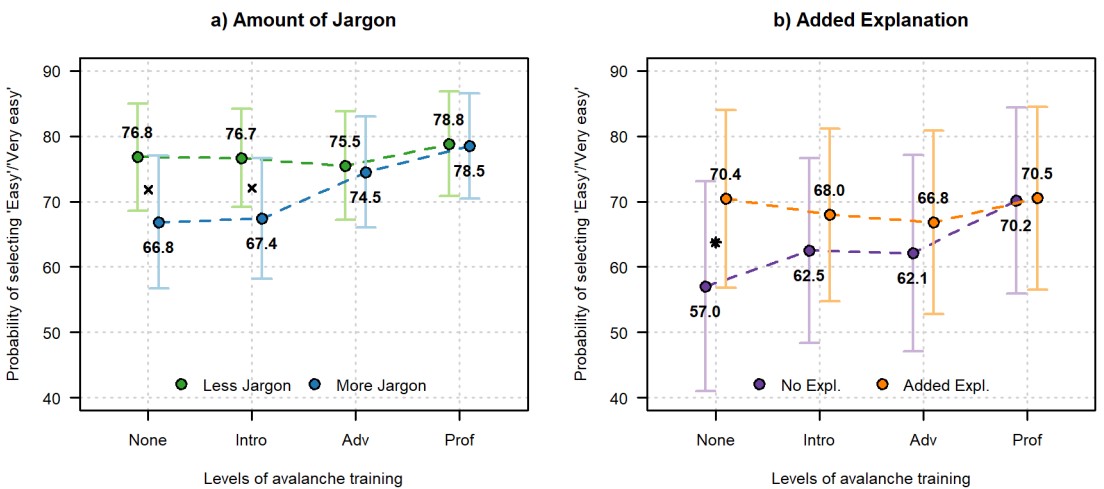

**Figure 6: Estimated marginal probabilities for selecting 'Easy' or 'Very Easy' for understandability of statement as function of a) the interaction effect of avalanche training and amount of jargon, b) the interaction effect of avalanche training and added explanation. Error bars represent 95% confidence intervals for probabilities calculated from the subsample for the particular parameter level. Significant post-hoc pairwise comparisons are indicated with asterisks (p-values < 0.01), diagonal crosses (0.01 ≤ p-values < 0.05) or horizontal crosses (0.05 ≤ p-values < 0.1). Variable levels for estimated marginal probabilities calculations: bulletin user type: Type D; avalanche awareness training: introductory; year of experience: 6-10 years; average days in backcountry per winter: 11-20 days; and attention to travel advice: considerable.**

As indicated by the random effects in Tables B3, we observed greater unexplained variance with individual participants than with the statements used for the jargon variations. This indicates that which specific statements participant saw had a smaller impact on their responses than variations in the characteristics of participants that the model did not account for. This gives us confidence that the specific selection of statements used did not unduly impact our results. In contrast, the variance of the statements with the explanation types was close to the level of variation for individual participants. Similar to the usefulness

models, statement type did not emerge as a significant predictor of participants' level of understanding ratings and was therefore removed from the analysis during the development of the models.

## 3.5 Recognition Confidence of Key Features in the Field

Out of the eighteen pairs of statements included in the analysis, seven of them referenced a specific terrain feature or snow condition resulting in 3,336 ratings of confidence recognizing a condition in the field. Hence, this dataset is therefore less than half the size of the dataset of the previous analysis. Approximately one third (33.8%) of participants who saw statements in this category reported that they would be fairly confident recognizing them in the field, and another third (33.8%) indicated that they would be very confident recognizing them in the field (scale: 'Not at all confident', 'Somewhat confident', 'Fairly confident', 'Very confident', and 'Extremely confident'). Only 3.3% of the ratings were 'Not at all confident'.

Similar to our approach in the other analyses, we built separate ordinal regression models to examine what factors contribute to the confidence ratings for the jargon and explanation samples separately. For the jargon sample, which consisted of 2,921 ratings for six pairs of statements (Table A1), we used an ordinal mixed regression model with participant ID, statement ID and version code as random effects. Our final jargon model included five predictor (bulletin user type, avalanche training, statement type, years of experience in the winter backcountry, and typical days per winter spent in the backcountry) as main effects and one interaction effect between statement type and avalanche training (Table B4). We subsequently built the model for the explanation sample (415 ratings) using the same predictor variables to ensure comparability of the results. However, since only one of our TTA statements with the explanation treatment included the recognition question (Table A1; Statement #8: "Watch for areas of hard wind slab on alpine features." vs. "Watch for areas of hard wind slab on alpine features. A good indicator is when travel suddenly gets easier because you do not sink in as much."), we did not need a mixed effects model and only estimated a standard ordinal regression model for this part of the analysis. Despite the much smaller sample size, all the included main and interaction effects emerged as significant (Table B4). However, it is important to note that the results of the explanation model lack generalizability because they only represent participants' perspective on a single statement. For the interested reader, the precise model specifications and R code for this analysis are also available in Haegeli et al. (2022).

As in the jargon model for understanding, participants' confidence ratings for identifying a particular feature in the field were partially driven by the level of jargon in a statement, participants' avalanche awareness training levels and the interaction between these two variables (Table B4). The resulting response pattern is nicely illustrated by the estimated marginal probabilities[3] for selecting 'Very confident' or 'Extremely confident' shown in Figure 7a. Overall, we see a strong increase in confidence with higher levels of training, but this effect is modulated by the amount of jargon. The effect is most pronounced for participants with introductory level training, who only had a 31.0% chance of being 'Very confident' or 'Extremely confident' if they saw a statement with higher levels of jargon, but it rose to 45.0% when they saw the version of the statement

---

[3] Estimated marginal probabilities for the confidence of recognition models were calculated using the following parameter levels: bulletin user type: Type D; avalanche awareness training: introductory; year of experience: 6-10 years; and average days in backcountry per winter: 11-20 days.

with lower levels of jargon (p-value = 0.0157). While the improvements for participants with no training was still substantial (30.3% vs. 40.7%), it was not large enough to be statistically significant (p-value = 0.1125). The improvements further diminished for participants with advanced and professional level training (p-values = 0.3933 and 0.469 respectively).

The explanation model revealed a similar pattern (Figure 7b) with the additional explanation significantly increasing the confidence of participants with no avalanche awareness training (46.9 vs. 17.3%; p-value = 0.0020) and introductory level training (38.0% vs. 23.6%; p-value = 0.0183). This is understandable since the explanation that was added to the statement provided a tip for how to recognize wind slabs. The confidence ratings of participants with higher levels of training were not affected by the added explanation. Even though the orange curve for the confidence ratings with the added explanation has a distinct u-shape, it is important to recognize the values do not differ significantly from each other. This is different from the jargon model (Figure 7a) where participants with professional level training rate their confidence significantly higher than other participants regardless of the level of jargon (p-values < 0.0001 and 0.0066, respectively).

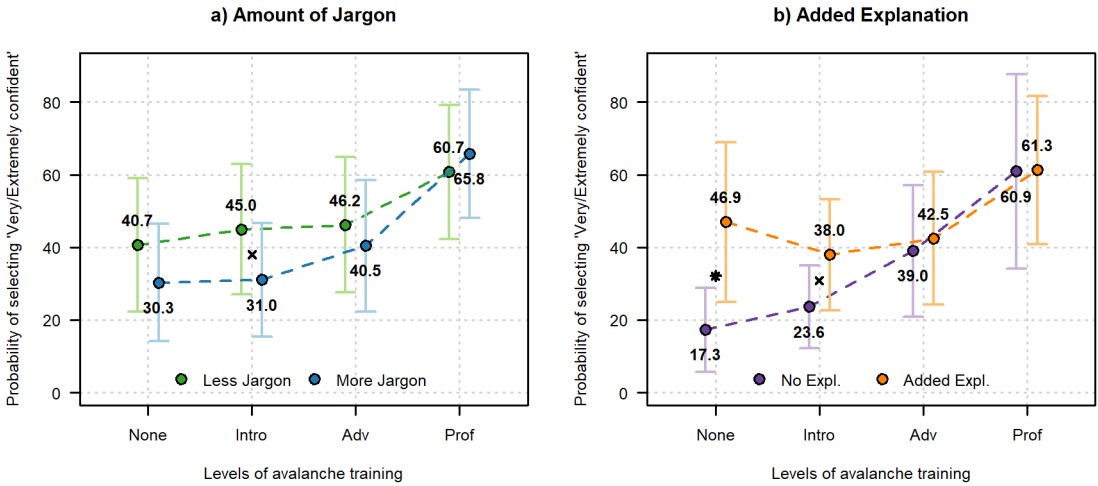

Figure 7: Estimated marginal probabilities for selecting 'Very confident' or 'Extremely confident' for recognizing condition in the field as function of a) the interaction effect of avalanche training and amount of jargon and b) the interaction effect of avalanche training and added explanation. Error bars represent 95% confidence intervals for probabilities calculated from the subsample for the particular parameter level. Significant post-hoc pairwise comparisons are indicated with asterisks (p-values < 0.01), diagonal crosses (0.01 ≤ p-values < 0.05) or horizontal crosses (0.05 ≤ p-values < 0.1). Variable levels for estimated marginal probabilities calculations: bulletin user type: Type D; avalanche awareness training: introductory; year of experience: 6-10 years; and average days in backcountry per winter: 11-20 days.

The jargon model further revealed that users with more advanced bulletin user types were more confident in their ability to recognize the conditions of the statements in the field. Type D users were significantly less likely to select 'Very confident' or 'Extremely confident' than Type E users (45.0% v. 54.9%, p-value = 0.0001) but significantly higher than Type C (31.2%, p-value < 0.0001). Type C participants did not differ significantly from Type Bs (24.9%, p-value = 0.3184).

Another predictor affecting participants' confidence level was their years of winter backcountry experience. Overall, participants with more years of experience were more likely to express that they were at least 'Very confident' (Figure 5b). Participants in their first year were the least likely to report that they were at least 'very' confident (25.5%), and the estimated mean probabilities continuously increase to 56.1% for participants with more than 20 years of winter backcountry experience. While this is the expected pattern, it is interesting to compare it to the effect experience has on participants' level of understanding ratings (Figure 5a). Even though the two rating scales are not directly comparable, it is interesting that the level of understanding ratings are overall at a higher level and level out more quickly than the confidence ratings that continue to increase with more years of experience. This difference in patterns was further highlighted when the respective models were estimated with years of experience included as polynomials instead of dummy coded, and the effect of the quadratic term being larger in the understandability model (not shown here; see Haegeli et al (2022) for details).

A final predictor that increased the likelihood of participants being confident in their ability to recognize a highlighted condition in the field was how many days they spent in the backcountry each winter (Table B4). As expected, more days tended to increase the likelihood that participants would have confidence in recognizing the condition. Not overly surprising, attention to TTA statements in general did not emerge as a significant predictor of confidence in recognizing a condition in the field.

Overall, the patterns in the parameter estimates of the standard ordinal regression model for the explanation treatment dataset were very similar to the jargon analysis, with the exception that the effect of the number of years of backcountry experience was more variable and did not exhibit the expected monotonous growth.

Similar to the models in the other jargon analyses, there was greater variance associated with individual participants than with the statements used, which means that the unaccounted variability among participants was bigger than the unaccounted variations among the TTA statements. As mentioned for the other models, this gives us confidence that the specific selection of statements used did not unduly impact our results. However, it is important to remember that the explanation model was based on a single statement. Furthermore, statement type did again not emerge as a significant predictor of participants' recognition confidence and was therefore removed during the development of the model.

## 4 Discussion

In this study we examined who is paying attention to the travel and terrain advice (TTA) statements in the bulletin, how useful participants find the advice, and if modifications to the advice could make it more useful for participants. We will describe key factors driving the responses to these questions and provide recommendations for avalanche warning services to optimize their TTA in avalanche bulletins.

### 4.1 Who is Paying Attention to Travel and Terrain Advice?

The TTA statements in an avalanche bulletin represents information that can help recreationists develop a risk management plan by guiding them towards appropriate terrain selection based on current avalanche hazard. Understanding who is using

this section of the bulletin allows avalanche warning services to identify which users incorporate this advice as part of their risk management process.

Significant patterns in who pays attention to the TTA emerged based on participants' bulletin user type, training, experience in the backcountry, and country of residence. Participants self-identifying as bulletin user Type D reported paying the most attention to the TTA statements included in the avalanche bulletin, followed by Type E. Types B and C paid significantly less attention than Types D and E, and too few Type A's responded to the survey to be included in the dataset. In the bulletin user typology, Type D bulletin users are characterized by their use of the information on the location and nature of specific

avalanche problems as part of their risk management approach for determining their trip objectives (St. Clair, 2019; St. Clair et al., 2021). It is therefore not surprising, that we see Type D users paying the greatest amount of attention to the TTA, which is the section of the bulletin explicitly targeted towards helping users develop a plan for how to travel through the terrain under the existing conditions. In contrast, Type C users make their travel decisions by 'opening and closing' avalanche terrain at a larger scale (St. Clair, 2019), and so the drop in attention we observe in this group may be because they tend not to incorporate

the specific terrain features described in the TTA into their risk management approach. The alignment of our results with predictions based of the bulletin user typology show that the TTA statements are being incorporated as expected as part of the risk management plan of users who incorporate specific terrain features into their analysis. To support these users, the information contained in the TTA should continue to highlight relevant slope-scale terrain features.

Additionally, after controlling for the bulletin user type, we also see a relationship between the personal experiences of

participants and the level of attention they pay to the TTA statements. Both higher levels of avalanche training and more years of experience in the backcountry lead to lower levels of attention to the TTA included in the bulletin. Participants with professional level training are significantly less likely to pay attention to the TTA than participants with lower levels of training, and there is a decreasing linear trend between the years of experience a participant has and their attention to the TTA. This pattern is not surprising, because more advanced users are more likely to already know the information conveyed in the TTA

based on their understanding of the avalanche problem information. These relationships demonstrate that it is less trained and less experienced users who are using the TTA advice, which makes it important to ensure that the advice is targeted towards these groups and is useful to them.

Finally, participants residing in the United States indicated higher levels of attention to the TTA than Canadian residents. While the results of our study are unable to provide specific insight on the reasons for this difference, we hypothesize that it

may be related to differences in avalanche bulletin format or outreach efforts. Many US-based avalanche bulletins integrate TTA statements as part of a prominent "bottom line" section, whereas Canadian avalanche warning services have historically had the TTA advice in the avalanche problem section on a secondary tab of their bulletins. While Canadian bulletins have recently moved the TTA advice to the front page of the bulletin, it is possible that user habits have not caught up with the change. It is also possible that differences in presentation of the TTA statements, such as including explanatory photos in US-

based bulletins, may lead to higher use by US residents. Further study is necessary to properly identify reasons for the difference between user attention to the TTA advice between participants located in Canada versus the United States.

Our results demonstrate that users who are integrating terrain into their daily planning but have lower levels of training or experience to support that integration are the current users of the TTA statements in bulletins. Hence, avalanche warning services should target the messaging of the TTA to the needs of these groups. Our findings suggest that the TTA is underused by participants who do not integrate terrain as part of their bulletin use, as well as participants who take advanced risk management approaches. Avalanche warning services can use this information to determine if additional products or information could be developed to better fit the needs of these user groups. In addition, the observed differences between Canadian and US participants should prompt additional communication between US and Canadian avalanche warning services to identify successful strategies for reaching more users in Canada.

## 4.2 What Determines the Usefulness of a Travel and Terrain Advice Statement?

With a better understanding of who is using the TTA statements, we turned towards investigating what makes TTA statements useful for users. In this section, we describe the factors that predict the usefulness of the TTA statements and how we interpret these factors.

### 4.2.1 Understanding and Recognition Confidence drive Usefulness

Participants' level of understanding and their confidence in recognition of the TTA statements both had a strong influence on how useful participants found the TTA statements. Higher levels of understanding and recognition confidence both led to higher usefulness ratings, and the range of the parameter estimates shows that participants' understanding of the advice is the more dominant of the two in determining the usefulness of the statements.

Our additional regression analyses allowed us to further investigate what contributes to these two main factors determining the usefulness of TTA statements. The regression models for how well participants understood the statements indicated that increases in the bulletin user type, level of training, years of experience, days spent recreating in the backcountry, and how much attention they pay to the TTA all increased the chances that participants would find the statements easier to understand. These same factors predicted how participants rated their recognition confidence, with the exception of how much attention they pay to TTA. The increase in both understanding and recognition confidence with additional training, experience, and a more sophisticated approach to the risk management is expected, as these are skills that develop over time and are taught as part of formal avalanche safety training courses. The absence of the attention to travel advice as a predictor for recognition confidence is also not surprising, as recognizing field conditions is not a bulletin-based skill and need to be developed through other channels.

Within these overall trends, there are some interesting differences in the predictors appeared in the models for both understanding and recognition confidence. The ratings of understanding increased quickly for participants with more than one year of experience but leveled off with no further differences between users with additional experience. In contrast, recognition confidence increased more gradually after the first year, and confidence continued to increase with additional years of experience. This suggests that confidence in recognizing conditions in the field develops more slowly than understanding does.

Recognizing specific terrain features or hazardous conditions is more difficult than simply understanding the phrases in the bulletin. This finding echoes the gap between comprehension and application of avalanche safety information among recreationists identified by Finn (2020). Most importantly, it highlights a need for continued opportunities for improvements in the application of the information provided in the bulletin, both during trip planning at home and in field. Future research into strategies to develop better terrain feature recognition, such as the inclusion of visual aids along with the TTA, should be considered to help users build their confidence in recognizing field conditions mentioned in the TTA.

The strong influence of understanding and recognition confidence on overall usefulness of the statements is important because it means that variations in these factors will also indirectly influence how useful the TTA statements are. By understanding what drives these additional variables, we are able to see more clearly how participants relate to the TTA statements. Our analyses show that users with less training and less experience are more likely to struggle with both understanding TTA statements and at recognizing the specific conditions mentioned in these statements. This should highlight to avalanche warning services and the avalanche safety community in general that more effort in education and skill building is needed for these groups of users.

### 4.2.2 Strong Links between Attention, Usefulness, and Understanding

In addition to ease of understanding and confidence in recognition, the amount of attention participants pay to the TTA statements was a significant predictor of how useful they find the statements, as well as how well they understand them. One possible way to interpret this result is that the amount of attention participants pay to the TTA represents their bulletin use practice similar to the avalanche bulletin user types described by St. Clair (2019). Bulletin users who pay more attention to the TTA statements might become more familiar with the terminology and messages over time and therefore find them more useful. This interpretation of the attention to TTA may also explain why bulletin user type did not emerge as a predictor in the usefulness model. Furthermore, it is consistent with the absence of this predictor in the recognition confidence model since recognizing a condition is a field-based skill and less tightly related to bulletin use practices.

Even though this interpretation seems intuitive, it is important to remember that regression analyses can only highlight association and not determine causation, and a reasonable alternative interpretation of the observed relationship could be that bulletin users pay more to the TTA statements because they find the statements more useful. However, since our survey presented each participant with a different subset of TTA statements, the structure of our dataset does not allow us to integrate participants' statement-specific usefulness and understanding ratings into the regression analysis for how much attention people pay to the TTA. Despite this limitation, our analysis highlights that the relationships between attention and usefulness, attention and understanding, and understanding and usefulness are strong and work together to drive user engagement with the TTA.

### 4.2.3 The Opposing Effects of Avalanche Training

While higher levels of avalanche training indirectly affect the usefulness of the TTA positively by leading to increased understanding and recognition confidence, the direct effect of training on the usefulness ratings turns out to be in the opposite direction. This means that at equal levels of understanding and recognition confidence, participants with higher levels of training perceive the TTA statements to be less useful, while participants with lower levels of training find the statements to be more useful. We interpret this result to indicate that while avalanche awareness training does increase one's understanding

of the TTA statement and confidence to recognize the described conditions in the field, participants with professional training may have the necessary avalanche risk management knowledge and skill to link avalanche hazard and terrain exposure without the explicit assistance provided by the TTA in the avalanche bulletin. This interpretation is consistent with the observation that the amount of attention to the TTA included in the bulletin decreases with increasing levels of avalanche awareness training. This highlights that the primary target audience for TTA statements are users with lower levels of training, and avalanche

warning services should seek to make sure the statements are optimized for these types of bulletin users

### 4.3 Can Travel and Terrain Advice Statements Be Made More Accessible to Users?

After controlling for all other factors, participants with the lowest levels of training found the TTA statements to be the most useful, but also demonstrated the lowest levels of understanding of the advice and the least confidence in recognizing the conditions in the field. This suggests that there may be a potential gap that these participants could be falling into, relying on

advice they do not completely understand. To close this gap, we tested two types of modifications to TTA statements to see if they could help to improve the understandability, recognition confidence, and overall usefulness of the statements.

### 4.3.1 Removal of Jargon

Simply removing the jargon from the TTA statements was enough to increase understanding of the statements among participants with no or introductory-level training to the same level as participants with advanced- or professional-level

training. Lowering jargon was also sufficient to boost the confidence in recognizing a condition in the field for participants with introductory-level training. As both understanding and recognition confidence are strong predictors of how useful participants find the TTA, it means that simply changing the phrasing of the statements will allow participants with low levels of training to make better use of the TTA without diminishing their clarity for users with more advanced training. This effect has been well documented in the science education and medical communities (e.g., Thomas et al., 2014; Bullock et al., 2019;

Rau et al., 2020). Studies on both cardiac patients and parents undergoing pre-natal counselling have identified that terms commonly used by professionals are not widely understood by patients, despite having visited these professionals (Thomas et al. 2014, Rau et al. 2020). Furthermore, Bullock et al. (2019) demonstrated that jargon reduces the ability to process scientific information and even impacts willingness to consider alternative perspectives or adopt new technologies. These studies are

important for the avalanche community because it is important that readers of TTA be able to both process the information as well as be open to adjusting their terrain exposure based on information within the TTA.

Interestingly, the lower levels of jargon did not affect the usefulness of the TTA statements beyond the indirect effects captured within the models for understanding and recognition confidence. We interpret this to mean that jargon is hard for users to interpret, but once the wording has been changed, it does not further affect the usefulness of the message of the advice given in the statement.

In this study, the removal of jargon had no effect on professional or advanced level users in any of the models. However, other studies express nuances in how jargon is perceived among laypeople. Zimmerman and Jucks (2018) showed that increased jargon impacts professional credibility both positively and negatively depending on the target audience for the communication. Their study emphasized that it is important to match the level of jargon to the intended audience of communication efforts. In the case of the avalanche bulletin, this supports our finding that jargon should be reduced in the TTA statements used by less advanced recreationists. However, it also implies that some jargon can still be used to communicate more precisely in messages targeted towards more advanced users, such as the snowpack and avalanche activity sections of the bulletin.

### 4.3.2 Added Explanation

In contrast to jargon, which only impacted usefulness via understanding and recognition confidence, adding additional explanations to the statements directly impacted how useful participants found the statements. Participants with introductory and advanced recreational training tended to find TTA statements with added explanations significantly more useful. The additional explanations provided information on context, how to identify the features, or the impacts of certain conditions (e.g., "Watch out for changes in the weather and snow conditions, *they may increase avalanche hazard as the day progresses*", or "Use extra caution around cornices: they are large, fragile, *and can trigger slabs on slopes below.").*

This increase in usefulness with the added explanation has also been observed in hurricane evacuation messaging research. The experimental study of Morss et al. (2016) demonstrated that warning messages that explained the potential impacts of an approaching hurricane have a bigger impact on participants' intentions to evacuate than messages without that added explanation. Additional work has refined the importance of these types of additions to forecasts by making the distinction between fear-based and impact-based messages. (Morss et al., 2018). In a study of individuals affected by Hurricane Sandy, four warning messages were trialed to determine how participants responded, including messages using non-personalized language to describe the impact of the storm, and messages using personalized language to trigger a fear-based reaction. In that study, high impact messages led to high evacuation intentions and higher risk perceptions than the fear-based message. Furthermore, the high-impact message was less likely to be perceived as overblown. From this, the authors concluded that adding impact messages that do not instill fear may have advantages. While our study did not investigate the role of fear-based messages, we suspect that the results of Morss et al. (2018) also apply to TTA statements. Given that backcountry recreationists voluntarily expose themselves to avalanche risk, including more information about the impacts of conditions in TTA

statements is likely even more useful to participants than fear-based messaging, which may lead to warning fatigue and loss of credibility.

Despite higher observed ratings among participants with no training or introductory training, added explanation did not significantly increase understanding or recognition confidence in participants. However, the effect was nearly significant, and a larger sample size may be sufficient to make the observed differences significant or allowed additional variables to emerge, particularly in the recognition confidence model where the sample size was reduced due to fewer questions.

### 4.4 Limitations

The participant sample in this study demonstrates trends consistent with previous surveys of backcountry recreation users. A high proportion of university educated, male, backcountry skiers, between 25 and 34 years of age with basic avalanche education engage in online surveys about avalanche safety (Finn, 2020; Haegeli and Strong-Cvetich, 2020; Haegeli et al. 2012). The similarity in sample demographics may be drawn from the similar survey promotion techniques used between this study and Finn (2020). Although this study and Finn (2020) was able to reach a wider range of users than previous studies, it only captures the behaviour of the demographic that responds to an online survey and may underrepresent non-English speaking participants or other demographics. Additional outreach efforts to underrepresented communities and communities that do not currently engage with online avalanche safety products are necessary to capture their effects on the wider backcountry winter recreation community. Even though the survey was open to all winter backcountry recreationists, the majority of participants were backcountry skiers, and the TTA statements were designed primarily from the perspective of backcountry skiers. Future studies should test if tailoring the statements for different activity groups, such as snowmobilers, snowshoers, or ice climbers, leads to improved usefulness of the statements for these users.

Our study also relies on self-reported metrics of understanding, recognition confidence, and usefulness. We did not include knowledge-based questions to test participant understanding and did not include field studies to determine if participants' confidence in their ability to recognize conditions in the field is warranted or not. The goal of this study was to understand how participants relate to the information provided in the bulletin, so while these self-reported metrics have limitations, we believe that they are appropriate for the objectives of this study. Future research may seek to understand how participants perceptions and self-reported ratings relate to their performance in field conditions.

Our study included a limited set of potential TTA statements, and the fact that the recognition confidence analysis of the explanation treatment only included a single statement is a significant limitation of that particular aspect of our study. However, our objective was to identify principles of communication via the TTA statements rather than suggest specific wording to warning services. Further research is needed to identify if additional trends in how the TTA is phrased, or if alternate coding of 'statement type' could lead to further insight into the usefulness of the TTA. We recommend that warning services work with members of the intended target audience to explicitly test the clarity and usefulness of their own specific TTA statements.

## 5 Conclusion

Selecting appropriate terrain while exposed to avalanche hazard is necessary to mitigate the risk of avalanches while traveling in the winter backcountry. While avalanche bulletins mainly focus on describing the hazard conditions, many of them also

provide travel and terrain advice (TTA) statements to help recreationists put the hazard information into action and navigate the backcountry safely. For this information to be effective, avalanche warning services need to understand who is using the advice, if the advice is useful to participants, and if altering the phrasing of the advice could broaden the accessibility of the information for more users. In this study, we identified that the core audience of the TTA in avalanche bulletins is users with introductory level avalanche awareness training who integrate slope-scale terrain considerations into their risk management

decisions (i.e., Type D bulletin users). Our results also highlight that simple statement modifications can considerably enhance the value of the TTA statements for the identified target audience. First, reducing jargon helps increase participants' level of understanding, which in turn makes the statements more useful for a broader audience. Second, adding additional information to the TTA statements that gives additional context or explanation to help clarify the statements makes the statements more meaningful. Taken together these findings indicate that the TTA statements are valuable for participants, and that making small

changes to the presentation of the TTA advice can further increase the usefulness for a wider group of users.

Avalanche warning services can implement these findings by creating communication guidelines for forecasters writing TTA statements that reduce jargon and include additional context for the statements. By improving communication of the TTA, avalanche warnings services can strengthen their role in helping recreationists not only understand avalanche hazard, but also how to mitigate their exposure to the hazard.

*Code and data availability.* The data, code, and output for our analysis and the data and code for the figures and tables included in this paper are available at https://doi.org/10.17605/OSF.IO/ACZX5 (Haegeli et al., 2022).

*Author Contribution.* KF and PH designed and executed the study and prepared the original draft. PM assisted with data
analysis and reviewed the manuscript. All authors reviewed the manuscript prior to submission.

*Competing Interests.* PH is a member of the editorial board of *Natural Hazards and Earth System Science*. The authors have no other competing interests to declare.

*Acknowledgements.* The authors thank Avalanche Canada, the Colorado Avalanche Information Center, and the Northwest Avalanche Center for their input during the design of the survey and their promotion of the study among their communities. We also thank all of our survey participants without whose contribution this research would not be possible. Two anonymous reviewers provided constructive feedback that helped us improve the quality of the manuscript substantially. We also thank Sven Fuchs for the handling of our manuscript. We are grateful to the Coast Salish peoples including the Tsleil-Waututh

(səlilw̓ətaʔɫ), Kwikwetlem (k<sup>w</sup>ik<sup>w</sup>əƛ̓əm), Squamish (Sḵwx̱wú7mesh Úxwumixw) and Musqueam (x<sup>w</sup>məθk<sup>w</sup>əy̓əm) Nations, on whose traditional and unceded territories Simon Fraser University and our research program resides. This research was conducted across Canada and the United States, which include the traditional territories of many other Indigenous Peoples.

*Financial support:* Kathryn C. Fisher received funding for this project from the Government of Canada Social Sciences and
Humanities Research Council via a Joseph Armand Bombardier Canada Graduate Scholarship Masters (CGS M). Public avalanche safety research at Simon Fraser University Avalanche Research Program is further supported by Avalanche Canada and the Avalanche Canada Foundation.

This appendix includes all the pairs of travel and terrain advice statements used in the survey.

**Table A1: Travel and terrain advice statements used in survey**

| ID | Statement 1 | Statement 2 | Modification Treatment | Statement Type | Questions |
|---|---|---|---|---|---|
| 1 | **Investigate the bond of the recent snow** before committing to your line. | **Check how well the recent snow sticks to the old snow surface before** committing to your line. | Jargon | Action | Understanding, usefulness |
| 2 | **Minimize exposure** to steep, sun exposed slopes, especially when the solar radiation is strong. | **Spend as little time as possible on or under** steep, sun exposed slopes, especially when the sun feels strong. | Jargon | Action | Understanding, usefulness |
| 3 | Avoid **lee and cross-loaded slopes** at and above treeline. | Avoid **slopes where blowing snow tends to deposit** at and above treeline. | Jargon | Action | Understanding, recognition, usefulness |
| 4 | Choose gentle slopes without exposure to **overhead hazard**. | Choose gentle slopes without **steep terrain above**. | Jargon | Action | Understanding, recognition, usefulness |
| 5 | In areas where deep persistent slabs may exist, avoid **shallow or variable depth snowpack areas**. | In areas where deep persistent slabs may exist, avoid **slopes that have areas where the snowpack is thinner**. | Jargon | Action | Understanding, recognition, usefulness |
| 6 | Avoid freshly **wind loaded features**, especially near ridge crests, roll-overs and in steep terrain. | Avoid **areas where blowing snow tends to deposit**, especially near ridge crests, roll-overs and in steep terrain. | Jargon | Action | Understanding, recognition, usefulness |
| 7 | Watch for areas of hard wind slab on **alpine features**. | Watch for wind slabs in **open areas at treeline and above**. | Jargon | Attitude | Understanding, recognition, usefulness |
| 8 | Watch for areas of **hard wind slab** on alpine features. | Watch for areas of **hard wind slab** on alpine features. A good indicator is when travel suddenly gets easier because you do not sink in as much. | Explanation | Attitude | Understanding, recognition, usefulness |
| 9 | Be aware of the potential for **remote triggering very large avalanches**. | Be aware of the potential for **triggering very large avalanches from flat areas that are typically not threatened by avalanches**. | Jargon | Attitude | Understanding, usefulness |
| 10 | **Use extra caution** around cornices: they are large, fragile and can trigger slabs on slopes below | **Use extra caution** around cornices: theses overhanging drifts of snow along ridge lines are large, fragile and can trigger slabs on slopes below. | Explanation | Attitude | Understanding, usefulness |

| | | | | |
|---|---|---|---|---|
| 11 | **Use caution** when approaching steep and rocky terrain. | **Use caution** when approaching steep and rocky terrain where even small avalanches might have severe consequences. | Explanation | Attitude | Understanding, usefulness |
| 12 | Remember that in the spring strong solar radiation and warm temperatures can weaken the snow in a matter of minutes. | Remember that in the spring strong solar radiation and warm temperatures can weaken the snow in a matter of minutes and make avalanche more likely. | Explanation | Attitude | Usefulness |
| 13 | **Watch out** for changes in the weather and snow conditions. | **Watch out** for changes in the weather and snow conditions because they may increase avalanche hazard as the day progresses. | Explanation | Attitude | Understanding, usefulness |
| 14 | Firm cornices can **pull back into flat terrain** at ridgetop if they fail. | Firm cornices can **pull back into flat terrain** at ridgetop if they fail. Some clear signs that you are on solid ground include the presence of trees, rocks. | Explanation | Fact | Understanding, usefulness |
| 15 | Recent new snow may be hiding windslabs that were easily visible before the snow fell. | Recent new snow may be hiding windslabs that were easily visible before the snow fell making it more difficult to recognize and avoid the avalanche problem. | Explanation | Fact | Usefulness |
| 16 | When a **thick melt-freeze surface crust** is present, avalanche activity is unlikely. | A **thick layer (15 cm or more) of frozen snow on the surface** is a good sign that avalanches are unlikely. | Jargon | Fact | Understanding, recognition, usefulness |
| 17 | The trees are currently not a **safe-haven**. | Staying in the trees is currently not a **good strategy for avoiding avalanches**. | Jargon | Fact | Understanding, usefulness |
| 18 | If triggered, storm slabs in-motion may **step down to deeper layers** and result in very large avalanches. | If triggered, small storm slabs may **trigger deeper layers** and cause very large avalanches. | Jargon | Fact | Understanding, usefulness |

## Appendix B

This appendix includes all parameter estimates and summary statistics for the models presented in this study.

**Table B1: Parameter estimates of regression model examining the attention paid to the TTA statements (Number of obs.: 2,998). Bolded levels are used to calculate the estimated marginal probabilities for the post-hoc tests.**

| Fixed Effects | | Parameter Estimate | Standard Error | p-value |
|---|---|---|---|---|
| **Main effects** | | | | |
| *Predictor* | *Level* | | | |
| Bulletin User Type | B | - | - | - |
| | C | -0.0340 | 0.1772 | 0.8476 |
| | **D** | 0.05858 | 0.1725 | 0.0007 |
| | E | 0.3904 | 0.1707 | 0.0222 |
| Avalanche training | None | - | - | - |
| | **Introductory** | 0.1146 | 0.1045 | 0.2724 |
| | Advanced | -0.0847 | 0.1265 | 0.5033 |
| | Professional | -0.532 | 0.1406 | 0.0002 |
| Days in backcountry/winter | Linear trend (**11-20 days**) | -0.1644 | 0.0386 | < 0.0001 |
| Country of residence | **Canada** | - | - | - |
| | United States | 0.3691 | 0.0796 | < 0.0001 |
| Intercept | None|Little | -5.298 | 0.3142 | < 0.0001 |
| | Little|Considerable | -2.4033 | 0.1999 | <0.0001 |
| | Considerable|Large | -0.0737 | 0.1922 | 0.07015 |

 **Table B2: Parameter estimates of the jargon and explanation models examining participants' usefulness ratings for the TTA statements. Bolded and italicized levels are used for calculating the estimated marginal probabilities in the post-hoc tests.**

| | | Jargon model (Number of obs.: 5,529) | | | Explanation model (Number of obs.: 3,371) | | |
|---|---|---|---|---|---|---|---|
| | | Parameter Estimate | Standard Error | p-value | Parameter Estimate | Standard Error | p-value |
| **Main effects** | | | | | | | |
| *Predictor* | *Level* | | | | | | |
| Attention to travel advice | Little | - | - | - | | | |
| | Some | 1.2883 | 0.1401 | <0.0001 | 1.0222 | 0.1424 | <0.0001 |
| | ***Considerable*** | 2.4247 | 0.1439 | <0.0001 | 1.8596 | 0.1464 | <0.0001 |
| Level of understanding | Very difficult & difficult | - | - | - | - | - | - |
| | Somewhat difficult | 1.5755 | 0.2356 | <0.0001 | 1.6516 | 0.2538 | <0.0001 |
| | Somewhat easy | 2.9044 | 0.2307 | <0.0001 | 2.8427 | 0.2517 | <0.0001 |
| | ***Easy*** | 4.1265 | 0.2350 | <0.0001 | 3.5723 | 0.2504 | <0.0001 |
| | Very easy | 5.5013 | 0.2449 | <0.0001 | 4.6015 | 0.2618 | <0.0001 |
| | Not applicable | n/a | | | 3.9645 | 0.4406 | <0.0001 |
| Recognition confidence | Not applicable | - | - | - | - | - | - |
| | Not at all | -2.6703 | 0.3149 | <0.0001 | -0.6164 | 0.7465 | 0.4090 |
| | Somewhat | -1.0238 | 0.2075 | <0.0001 | 0.0671 | 0.5351 | 0.9002 |
| | ***Fairly*** | -0.1070 | 0.1859 | 0.56492 | 0.6931 | 0.4961 | 0.1624 |
| | Very | 0.2235 | 0.1860 | 0.22951 | 0.7896 | 0.5019 | 0.1157 |
| | Extremely | 0.6885 | 0.2128 | 0.00121 | 1.2187 | 0.5539 | 0.0278 |
| Avalanche training | None | - | - | - | - | - | - |
| | ***Introductory*** | -0.16761 | 0.12667 | 0.18577 | -0.1443 | 0.1419 | 0.3094 |
| | Advanced | -0.1267 | 0.1512 | 0.40196 | -0.1711 | 0.1685 | 0.3097 |
| | Professional | -0.4683 | 0.1626 | 0.00398 | 0.0832 | 0.1867 | 0.6558 |
| Statement treatment | ***Less jargon*** | - | - | - | n/a | | |
| | More jargon | 0.0358 | 0.1639 | 0.82718 | n/a | | |
| | ***No explanation*** | n/a | | | - | - | - |
| | Added explanation | n/a | | | 0.3318 | 0.1846 | 0.0723 |
| **Interaction effects** | | | | | | | |
| *Predictor (level)* | *Predictor (level)* | | | | | | |
| Statement treatment | Avalanche Training | | | | | | |
| More Jargon[a] | None | - | - | - | n/a | | |
| | Introductory | -0.0467 | 0.1627 | 0.77392 | n/a | | |
| | Advanced | -0.1215 | 0.1928 | 0.52874 | n/a | | |

| | | Parameter Estimate | Standard Error | p-value | Parameter Estimate | Standard Error | p-value |
|---|---|---|---|---|---|---|---|
| | Professional | -0.0903 | 0.2048 | 0.65916 | n/a | | |
| Added explanation[b] | None | n/a | | | - | - | - |
| | Introductory | n/a | | | 0.2160 | 0.1941 | 0.2658 |
| | Advanced | n/a | | | -0.0069 | 0.2267 | 0.9756 |
| | Professional | n/a | | | -0.3697 | 0.2484 | 0.1366 |

| Threshold | Parameter Estimate | Standard Error | p-value | Parameter Estimate | Standard Error | p-value |
|---|---|---|---|---|---|---|
| Not at all \| Somewhat | -0.5693 | 0.2986 | 0.0566 | 0.8982 | 0.3430 | 0.0088 |
| Somewhat \| Fairly | 2.8148 | 0.3037 | <0.0001 | 3.4799 | 0.3578 | <0.0001 |
| Fairly \| Very | 5.0964 | 0.3149 | <0.0001 | 5.2358 | 0.3723 | <0.0001 |
| Very \| Extremely | 8.3283 | 0.3387 | <0.0001 | 7.6587 | 0.3985 | <0.0001 |

| Random Effects | Number of groups | Variance | Standard Deviation | Number of groups | Variance | Standard Deviation |
|---|---|---|---|---|---|---|
| Participant ID | 2966 | 1.4508 | 1.2045 | 2482 | 0.6275 | 0.7921 |
| Version Code : Statement ID | 22 | 0.0404 | 0.2009 | 14 | 0.0221 | 0.1486 |
| Statement ID | 11 | 0.0541 | 0.2326 | 7 | 0.1630 | 0.4038 |

[a] Base level is 'Less jargon'.
[b] Base level is 'No explanation'.

**Table B3: Parameter estimates of the jargon and explanation models examining participants' ease of understanding of the TTA statements. Bolded and italicized levels are used for calculating the estimated marginal probabilities for the post-hoc tests.**

| | | Jargon model (Number of obs.: 5,529) | | | Explanation model (Number of obs.: 2,290) | | |
|---|---|---|---|---|---|---|---|
| | | Parameter Estimate | Standard Error | p-value | Parameter Estimate | Standard Error | p-value |
| **Main effects** | | | | | | | |
| *Predictor* | *Level* | | | | | | |
| Bulletin user type | B | - | - | - | - | - | - |
| | C | 0.3287 | 0.1685 | 0.0511 | 0.0246 | 0.2105 | 0.9069 |
| | ***D*** | 0.7114 | 0.1639 | <0.0001 | 0.4659 | 0.2052 | 0.0232 |
| | E | 1.0097 | 0.1634 | <0.0001 | 0.5103 | 0.2033 | 0.0121 |
| Avalanche training | None | - | - | - | - | - | - |
| | ***Introductory*** | -0.0071 | 0.1254 | 0.9549 | 0.2276 | 0.1630 | 0.1625 |
| | Advanced | -0.0728 | 0.1528 | 0.6338 | 0.2115 | 0.1986 | 0.2870 |
| | Professional | 0.1170 | 0.1689 | 0.4887 | 0.5731 | 0.2265 | 0.0114 |
| Days in backcountry/winter | 1-10 days | - | - | - | - | - | - |
| | ***11-20 days*** | 0.1120 | 0.0993 | 0.2595 | 0.1950 | 0.1213 | 0.1080 |
| | 21-50 days | 0.3222 | 0.1012 | 0.0015 | 0.4294 | 0.1229 | 0.0005 |
| | 51+ days | 0.3672 | 0.1281 | 0.0042 | 0.4638 | 0.1557 | 0.0029 |
| Years of experience | First year | - | - | - | - | - | - |
| | 2-5 years | 0.4808 | 0.1718 | 0.0051 | 0.3732 | 0.2016 | 0.0642 |
| | ***6-10 years*** | 0.5312 | 0.1814 | 0.0034 | 0.3101 | 0.2141 | 0.1475 |
| | 11-20 years | 0.4840 | 0.1843 | 0.0086 | 0.2887 | 0.2174 | 0.1842 |
| | 21+ years | 0.5683 | 0.1814 | 0.0017 | 0.6022 | 0.2133 | 0.0048 |
| Attention to travel advice | Little | - | - | - | - | - | - |
| | Some | 0.1884 | 0.1323 | 0.1543 | 0.2959 | 0.1630 | 0.0694 |
| | ***Considerable*** | 0.5923 | 0.1313 | <0.0001 | 0.5166 | 0.1621 | 0.0014 |
| Statement treatment | ***Less jargon*** | - | - | - | n/a | | |
| | More jargon | -0.4960 | 0.2172 | 0.0224 | n/a | | |
| | ***No explanation*** | n/a | | | - | - | - |
| | Added explanation | n/a | | | 0.5854 | 0.2207 | 0.0080 |
| | | | | | | | |
| **Interaction effects** | | | | | | | |
| *Predictor (level)* | *Predictor (level)* | | | | | | |
| Statement treatment | Avalanche Training | | | | | | |
| More Jargon[a] | None | - | - | - | n/a | | |
| | Introductory | 0.0338 | 0.1560 | 0.8284 | n/a | | |

| | | Parameter Estimate | Standard Error | p-value | Parameter Estimate | Standard Error | p-value |
|---|---|---|---|---|---|---|---|
| | Advanced | 0.4434 | 0.1866 | 0.0175 | n/a | | |
| | Professional | 0.4790 | 0.2000 | 0.0166 | n/a | | |
| Added explanation[b] | None | n/a | | | - | - | - |
| | Introductory | n/a | | | -0.3431 | 0.2224 | 0.1229 |
| | Advanced | n/a | | | -0.3799 | 0.2636 | 0.1496 |
| | Professional | n/a | | | -0.5700 | 0.2941 | 0.0526 |

| Threshold | Parameter Estimate | Standard Error | p-value | Parameter Estimate | Standard Error | p-value |
|---|---|---|---|---|---|---|
| (Very) Diff \| SW Diff | -2.6825 | 0.3135 | <0.0001 | -1.7154 | 0.4137 | <0.0001 |
| SW Diff \| SW Easy | -1.1937 | 0.3062 | 0.0001 | -0.2323 | 0.4074 | 0.5686 |
| SW Easy \| Easy | 0.3459 | 0.3054 | 0.2572 | 0.9840 | 0.4081 | 0.0159 |
| Easy \| Very Easy | 2.5270 | 0.3095 | <0.0001 | 2.6398 | 0.4160 | <0.0001 |

| Random Effects | Number of groups | Variance | Standard Deviation | Number of groups | Variance | Standard Deviation |
|---|---|---|---|---|---|---|
| Participant ID | 2977 | 1.2661 | 1.1252 | 1954 | 0.3584 | 0.5987 |
| Version Code : Statement ID | 22 | 0.1611 | 0.4014 | 10 | 0.0315 | 0.1775 |
| Statement ID | 11 | 0.1656 | 0.4070 | 5 | 0.3166 | 0.5626 |

[a] Base level is 'Less jargon'.
[b] Base level is 'No explanation'.

**Table B4: Parameter estimates of the jargon and explanation models examining participants' confidence in recognizing the highlighted feature in the TTA statements. Bolded and italicized levels are used for calculating the estimated marginal probabilities for the post-hoc tests.**

| | | Jargon model (Number of obs.: 2,921) | | | Explanation model (Number of obs.: 415) | | |
|---|---|---|---|---|---|---|---|
| | | Parameter Estimate | Standard Error | p-value | Parameter Estimate | Standard Error | p-value |
| **Main effects** | | | | | | | |
| *Predictor* | *Level* | | | | | | |
| Bulletin user type | B | - | - | - | - | - | - |
| | C | 0.3098 | 0.2044 | 0.1296 | 1.2485 | 0.6091 | 0.0404 |
| | ***D*** | 0.9007 | 0.1976 | <0.0001 | 2.1887 | 0.6092 | 0.0003 |
| | E | 1.2975 | 0.1983 | <0.0001 | 2.7468 | 0.6104 | <0.0001 |
| Avalanche training | None | - | - | - | - | - | - |
| | ***Introductory*** | 0.1763 | 0.1567 | 0.2606 | 0.3924 | 0.3388 | 0.2468 |
| | Advanced | 0.2267 | 0.1867 | 0.2247 | 1.1195 | 0.4048 | 0.0057 |
| | Professional | 0.8143 | 0.2078 | 0.0001 | 2.0087 | 0.5807 | 0.0005 |
| Days in backcountry/winter | 1-10 days | - | - | - | - | - | - |
| | ***11-20 days*** | 0.5334 | 0.1210 | <0.0001 | 0.5831 | 0.2797 | 0.0371 |
| | 21-50 days | 0.8217 | 0.1244 | <0.0001 | 0.6945 | 0.2840 | 0.0145 |
| | 51+ days | 1.2870 | 0.1571 | <0.0001 | 1.0612 | 0.3478 | 0.0023 |
| Years of experience | First year | - | - | - | - | - | - |
| | 2-5 years | 0.6304 | 0.2106 | 0.0028 | 1.7405 | 0.4553 | 0.0001 |
| | ***6-10 years*** | 0.8709 | 0.2220 | 0.0001 | 1.3881 | 0.4749 | 0.0035 |
| | 11-20 years | 1.1498 | 0.2274 | <0.0001 | 2.2502 | 0.5037 | <0.0001 |
| | 21+ years | 1.3188 | 0.2238 | <0.0001 | 1.6593 | 0.4716 | 0.0004 |
| Statement treatment | ***Less jargon*** | - | - | - | n/a | | |
| | More jargon | -0.4547 | 0.2848 | 0.1104 | n/a | | |
| | ***No explanation*** | n/a | | | - | - | - |
| | Added explanation | n/a | | | 1.4429 | 0.4246 | 0.0008 |
| | | | | | | | |
| **Interaction effects** | | | | | | | |
| *Predictor (level)* | *Predictor (level)* | | | | | | |
| Statement treatment | Avalanche Training | | | | | | |
| More Jargon[a] | None | - | - | - | n/a | | |
| | Introductory | -0.1413 | 0.2154 | 0.5120 | n/a | | |
| | Advanced | 0.2197 | 0.2503 | 0.3800 | n/a | | |
| | Professional | 0.6716 | 0.2766 | 0.0152 | n/a | | |
| Added explanation[b] | None | n/a | | | - | - | - |
| | Introductory | n/a | | | -0.7600 | 0.5042 | 0.1317 |

| | Parameter Estimate | Standard Error | p-value | Parameter Estimate | Standard Error | p-value |
|---|---|---|---|---|---|---|
| Advanced | n/a | | | -1.2991 | 0.5796 | 0.0250 |
| Professional | n/a | | | -1.4270 | 0.7079 | 0.0438 |

| **Threshold** | **Parameter Estimate** | **Standard Error** | **p-value** | **Parameter Estimate** | **Standard Error** | **p-value** |
|---|---|---|---|---|---|---|
| Not at all \| Somewhat | -1.9329 | 0.4451 | <0.0001 | 1.1346 | 0.7242 | 0.1172 |
| Somewhat \| Fairly | 0.4198 | 0.4383 | 0.3381 | 3.4453 | 0.7494 | <0.0001 |
| Fairly \| Very | 2.6833 | 0.4464 | <0.0001 | 5.7252 | 0.7877 | <0.0001 |
| Very \| Extremely | 5.1407 | 0.4649 | <0.0001 | 7.7952 | 0.8091 | <0.0001 |

| **Random Effects** | **Number of groups** | **Variance** | **Standard Deviation** | **Number of groups** | **Variance** | **Standard Deviation** |
|---|---|---|---|---|---|---|
| Participant ID | 2252 | 0.8235 | 0.9075 | n/a | n/a | n/a |
| Version Code : Statement ID | 12 | 0.1414 | 0.3760 | n/a | n/a | n/a |
| Statement ID | 6 | 0.5630 | 0.7503 | n/a | n/a | n/a |

[a] Base level is 'Less jargon'.
[b] Base level is 'No explanation'.

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
