# Peer review of "Travel and terrain advice statements in public avalanche bulletins: A quantitative analysis of who uses this information, what makes it useful, and how it can be improved for users"

_Natural Hazards and Earth System Sciences, 2021_

## Referee Comment (RC1)

**General comments**

A nice manuscript. You nicely analyze the usefulness of travel and terrain advises (TTA) in avalanche reports. This study shows, that TTA's are most useful for a wide range of recreationists with an intermediate knowledge level. Novelists may find it helpful but cannot understand TTA's, and for experts it is an information which they have derived from the avalanche report anyway and therefore is not really necessary.

Your paper is well written, easy understandable and very concise. It adds a piece to the knowledge puzzle among avalanche warning services and gives them helpful advices to improve TTA's in their avalanche reports.

I propose hereby some minor revisions as listed below. Since I see this piece of work as relevant I encourage the authors to undertake the suggested revisions.

**Specific comments**

Line 25-28: It is true, that information on winter backcountry participation is sparse. However, one study shows at least numbers from Switzerland: (Winkler, Fischer, & Techel, 2016).

Line 46-49: I suggest to also mention the 'avalanche prone locations' and the 'danger description' in the information pyramid.

Line 86-94: Up to my knowledge in Europe, some regions/countries (e.g. Italy) have strong concerns using TTA due to legal consequences of such direct and clear advises. It would be nice, if you mention this aspect here.

Line 217: Is there any reason to exclude folks younger than 20 years old? If yes, could you mention it?

Line 220/221 and 223/24: years of backcountry experience is excluded twice.

Line 525/526: The effort in education is probably not only addressed to avalanche warning services but more general to avalanche prevention institutions.

Line 551/552: In addition, I guess that the TTA is too simple for avalanche professionals – it may guide them to limited. Avalanche professionals most probably derive a whole set of TTA's from the bulletin.

Limitations, Line 613 ff: In my eyes an important limitation of such online survey is, that one only reaches recreationists with a minimal awareness of avalanche risk management. People with no or very few avalanche awareness, who do not look at avalanche reports, are most probably not reached which such surveys. Or did you reach them by Social media probably more than in past? However, I guess there is a bias towards more educated recreationists. Anyway, it would be nice to give a bit more attention to this aspect.

**Technical notes**

Line 172: who instead of how?

**References**

Winkler, K., Fischer, A., & Techel, F. (2016). Avalanche risk in winter backcountry touring: Status and recent trends in Switzerland. In E. Greene (Ed.), *International Snow Science Workshop 2016, Proceedings* (pp. 270-276). Breckenridge, CO.

---

## Author Response (AR1)

April 27, 2022

Dear Dr. Fuchs:

It is with great pleasure that we are submitting the revised version of our manuscript *"Travel and terrain advice statements in public avalanche bulletins: A quantitative analysis of who uses this information, what makes it useful, and how it can be improved for users"* to NHESS for publication. We would like to thank the two reviewers for their constructive comments, and we really appreciated the two extensions you granted for the revision of the manuscript.

While the main conclusions of the paper remain unchanged, the analysis presented in the manuscript is considerably cleaner since the models for the effects of the reduced jargon and the added explanations have now been separated. Addressing the other reviewer comments further improved the manuscript.

The following pages describe our responses to the reviewer comments in detail. However, since the requested revisions required extensive revisions to the description of the analysis (Section 2.3) and the results section (Section 3), the manuscript version with track changes does not explicitly show every change, but rather highlights the sections that have been changed with author comments.

In addition to addressing the concerns of the reviewers, we also carefully edited the entire manuscript.

We hope that our revisions are to your satisfaction. Please let us know if you have any questions or required additional information.

On behalf of the authors

Pascal Haegeli, PhD
Simon Fraser University, Vancouver BC, Canada

*Fisher et al.: Travel and terrain advice statements in public avalanche bulletins: A quantitative analysis of who uses this information, what makes it useful, and how it can be improved for users*

**Response to Reviewer 1**

February 5, 2022

**Reviewer comment:** A nice manuscript. You nicely analyze the usefulness of travel and terrain advises (TTA) in avalanche reports. This study shows, that TTA's are most useful for a wide range of recreationists with an intermediate knowledge level. Novelists may find it helpful but cannot understand TTA's, and for experts it is an information which they have derived from the avalanche report anyway and therefore is not really necessary.

Your paper is well written, easy understandable and very concise. It adds a piece to the knowledge puzzle among avalanche warning services and gives them helpful advices to improve TTA's in their avalanche reports.

I propose hereby some minor revisions as listed below. Since I see this piece of work as relevant I encourage the authors to undertake the suggested revisions.

*Author response: We thank the reviewer for taking the time to review our manuscript and providing supportive and constructive comments.*

**Specific comments**

**1.1 Backcountry use numbers**

**Reviewer comment:** Line 25-28: It is true, that information on winter backcountry participation is sparse. However, one study shows at least numbers from Switzerland: (Winkler, Fischer, & Techel, 2016).

*Author response: We thank the reviewer for pointing out this reference. We added this reference in the second sentence of the introduction (p. 1).*

**1.2. Information pyramid**

**Reviewer comment:** Line 46-49: I suggest to also mention the 'avalanche prone locations' and the 'danger description' in the information pyramid.

*Author response: 'Avalanche prone locations' and 'danger descriptions' are labels of bulletin components that are presented in European avalanche bulletins. Since our study focuses on North American avalanche bulletins, mentioning Swiss or European avalanche bulletin components does not seem meaningful. However, to clarify this, we modified this section of our introduction as follows (end of second paragraph on p. 2):*

*"Reflecting this process, avalanche bulletins in North America present the avalanche hazard information to their readers in a pyramid-like structure with the overall hazard rating given first, then details of avalanche problems, and finally additional details about snowpack structure, avalanche observations, and weather conditions. Avalanche bulletins in Europe use a similar but slightly different structure (EAWS, 2021)."*

**1.3 Legal concerns**

**Reviewer comment:** Line 86-94: Up to my knowledge in Europe, some regions/countries (e.g. Italy) have strong concerns using TTA due to legal consequences of such direct and clear advises. It would be nice, if you mention this aspect here.

*Author response: We thank the reviewer for this background information. However, after careful consideration, we decided not to include this information into our paragraph on the presentation of TTA in different bulletins. There are two reasons for this. First, our description focuses on how the TTA is presented in different bulletins which relates to the main objective of our research. Exploring the reasons why TTA is included by some and not by others would be a different topic. More importantly, however, we do not have a meaningful reference for properly citing this information.*

**1.4 Participants younger than 20 years**

**Reviewer comment:** Line 217: Is there any reason to exclude folks younger than 20 years old? If yes, could you mention it?

*Author response: Participants younger than 20 were excluded from the analysis since that age category could include minors, and the survey did not allow us to get consent from a parent or legal guardian. We added a footnote explaining the situation on page 10.*

**1.5 Typo**

**Reviewer comment:** Line 220/221 and 223/24: years of backcountry experience is excluded twice.

*Author response: Thanks for pointing this out. We deleted the description of the exclusion of participants who did not provide information about their backcountry experience in the first sentence.*

**1.6 Target audience for recommendations**

**Reviewer comment:** Line 525/526: The effort in education is probably not only addressed to avalanche warning services but more general to avalanche prevention institutions.

*Author response: To highlight that our conclusion is relevant for the broader avalanche safety community, we expanded the last sentence in Section 4.2.1 (p. 25) as follows:*

*"This should highlight to avalanche warning services **and the avalanche safety community in general** that more effort in education and skill building is needed for these groups of users."*

**1.7 Usefulness of TTA to professionals**

**Reviewer comment:** Line 551/552: In addition, I guess that the TTA is too simple for avalanche professionals – it may guide them to limited. Avalanche professionals most probably derive a whole set of TTA's from the bulletin.

*Author response: We believe that professional guides have the skills to make appropriate terrain choices directly based on the hazard description and to not need the interpretation of the TTA. Since this is more of a personal comment than a concern, we do not think that it requires any modifications of the manuscript.*

**1.8 Limitations**

**Reviewer comment:** Limitations, Line 613 ff: In my eyes an important limitation of such online survey is, that one only reaches recreationists with a minimal awareness of avalanche risk management. People with no or very few avalanche awareness, who do not look at avalanche reports, are most probably not reached which such surveys. Or did you reach them by Social media probably more than in past? However, I guess there is a bias towards more educated recreationists. Anyway, it would be nice to give a bit more attention to this aspect.

**Author response:** *We completely agree with the reviewer that this is an inherent limitation of these types of surveys, and that people with limited avalanche awareness are difficult to engage in this type of research. In response to this comment, we expanded the first paragraph of our limitation section (Section 4.4; p. 28) to explain the potential biases in our sample in more detail, and included some recommendations about how these issues could be addressed.*

**1.9 Typo**

**Reviewer comment:** Line 172: who instead of how?

**Author response:** *We fixed this typo.*

**Response to Reviewer 2**

February 5, 2022

***Reviewer comment:*** The aim of this study is to evaluate the use and effectiveness of the travel and terrain advice (TTA) statements in daily avalanche bulletins. More specifically, the authors set out to identify which user groups that pay attention to the TTA, how useful different user groups find the TTA, and if the usefulness can be increased by minor changes in the phrasing of the message.

The study addresses an important question for avalanche warning services (AWS) and other natural hazards warning services, i.e., how can we make the public warnings more useful and understandable for the targeted population? The authors use a relatively large and heterogenous sample of backcountry recreationists (N = 3100). This increases the chances that the results are generalizable to the general population of users of the avalanche bulletin. The authors employ a multilevel ordinal regression model (mixed effects model). Since the outcome variables are ordinal, and since the data is grouped on both the individual and TTA statement level, this seems like a reasonable approach. However, as described in the comments below, I also think that the estimation procedure creates challenges that are not completely addressed by the authors. Finally, I really like that the authors use interaction effects, as this allows them to analyze differences in effects in different user groups.

In conclusion, I find that this paper presents research that represent a substantial contribution to the literature on risk communication, and therefore contributes to our understanding of natural hazards and their consequences (Scientific Significance: level 1). The method used is valid. However, I think that some robustness checks are needed and that the discussion of the results would benefit from some restructuring (Scientific Quality: level 2-3). The presentation quality is excellent (level 1).

***Author response:*** *We thank the reviewer for taking the time to study our manuscript in detail and provide such constructive comments. We particularly appreciate their suggestions for how to address the shortcomings and improve the analysis approach and writing.*

**Comments**

**2.1 Low number of Bulletin User Type As**

***Reviewer comment:*** In spite of the relatively large sample size, the distribution of responses across both attention to the TTA and avalanche bulletin user types is very skewed. Most importantly, only nineteen participants stated that they pay no attention to the TTA. Ordinal models are sensitive to the number of observations in each cell. The low number of observations in the first cell of attention to TTA may create problems. This problem is amplified by the fact that the distribution of participants in the different avalanche bulletin user types is also skewed. There are only 15 participants, who self-categorize as type "A". Of these, two pay no attention to the TTA.

The low number of type A bulletin users may cause problems in all models in the paper. You use type A as a reference group in your analyses. In other words, the other user types are compared to type A users and not to other types. You in general find that avalanche bulletin user type is an important predictor in all models. However, based on the results presented in the tables, it seems like the coefficients on type "B"-"F" are very similar, i.e., the main effect appears to be between "higher than A" and "A". Since there are only 15 participants of type "A", these individuals are given a disproportional weight.

Suggestions: 1) I would like to suggest that you check how well the models fit the data, e.g., by comparing the predicted probabilities in each cell (level of attention), with the actual distribution in the data. This is especially important for the TTA attention model. If you find that the model over-predicts observations in the lowest cell (no attention), I suggest combining this category with the next lowest category (little attention), and re-running your analysis on this variable.

*Author response: We appreciate the reviewer highlighting this challenge in our response variables. While the reviewer focused on the TTA attention model, this is an issue in all models as participants chose the lowest response categories only very rarely. To be more transparent about this, our revised manuscript includes information about the response variable distributions in the descriptions of all models. Interested readers can also directly examine the distributions using the data and code published at https://doi.org/10.17605/OSF.IO/ACZX5.*

*Note that low response categories in ordinal regression are not an issue per se. For instance, some researchers even advocate the use of ordinal regression for some metric responses (see, e.g., Section 15.3 of the influential regression book by Harrell, 2015). However, we understand the reviewer's concern.*

*One approach to demonstrate whether our models are able to meaningfully represent the observed distributions of the response categories in R is to use the posterior_predictive_check function in the posterior package, which "simulates replicated data under the fitted model and then compares these to the observed data" (Gelman and Hill, 2007, p. 158). However, since this function does not support polr or clmm models, we calculate the simulated response category frequencies by hand. Our results indicate that all of our models capture the observed frequencies nicely. Hence, there is no need to simplify the models by combining the lowest and least frequent response category with the next higher response category as suggested by the reviewer. However, we did combine 'Very difficult' and 'Difficult' in the understanding models due to the very small response frequencies (< 1%).*

*The revised manuscript describes our approach in detail in the fourth paragraph of the Data Analysis section (Section 2.3; p. 8-9). Furthermore, we describe the results of the posterior predictive checks (see code for details) in the results section for every model (see, e.g., last sentence of Section 3.2 on p. 11).*

*Harrell, Jr., F. E. (2015). Regression Modeling Strategies: With Applications to Linear Models, Logistic and Ordinal Regression, and Survival Analysis (2nd ed.). New York: Springer.*

*Gelman, A., & Hill, J. (2007). Data analysis using regression and multilevel/hierarchical models. Cambridge; New York: Cambridge University Press*

2) To evaluate if the difference between type "A" users and higher level users is driving the effect, I suggest two approaches: either combining type "A" with type "B", or run the model on the subsample of participants of type "B" or higher. This comment holds for all models in the paper. The robustness tests can be included in an online appendix if the results do not differ in a significant way.

*Author response: After careful consideration, we decided to eliminate participants who self-identified as Bulletin User Types A and F from all models in our analysis including the mixed effects models examining the ease of understanding, recognition in the field and overall usefulness. The sample sizes for these two bulletin user types are relatively small (A: 15; F: 87), and since the main focus of the study is on the effect of formal avalanche training, the two levels do not add much. Eliminating these two user types from the analysis also reduce the correlation between training and bulletin user type (from 0.407 to 0.350), which*

*partially addresses Reviewer Concern 2.2. The removal of bulletin user types A and F is now described inb the first paragraph of Section 3.1 (p. 10), which describes the characteristics of participants.*

*Overall, the main patterns of the analyses do not change with this slight adjustment of the sample.*

**2.2 Correlation between training and bulletin user type**

The correlation between avalanche training and avalanche bulletin user type is relatively high. For example, there are no participants of type "A" with advanced or professional training, and no participants of type "F" without professional training. This may bias the results. The correlation may cause problems with inflated variance, and you will have some cells with zero observations.

Suggestions: I recommend comparing three different models: One where you include both type and training, one with just training and one with just type, and compare the models in terms of goodness of fit.

*Author response: This correlation is expected as participants with higher levels of avalanche awareness are expected to use avalanche bulletins in a more sophisticated way. As described in our response to Reviewer Comment 2.1, we eliminated participants who self-reported as Bulletin User Type A and F from the analysis, which reduced this correlation across the entire analysis.*

*To explore the impact of the remaining correlation between these two predictors on the standard errors of the regression parameters, we initially compute the generalized variance inflation factor (GVIF; Fox & Monette, 1992) to assess whether there is any issue. Using the general rule of thumb for categorical variables that $GVIF^{(1/(2*DF))} < 5$ indicating no issues with collinearity with other variables, we can confirm that collinearity is generally not an issue in our models.*

*The GVIFs are now calculated for every model, and comments are included in the result section stating that the are no issues with correlations between predictor variables. See highlighted sections in track-changes version of manuscript (e.g., last sentence in first paragraph of Section 3.2).*

*Fox, J. and Monette, G. (1992) Generalized collinearity diagnostics. Journal of the American Statistical Association, 87, 178-183.*

*Also see http://web.vu.lt/mif/a.buteikis/wp-content/uploads/PE_Book/4-5-Multiple-collinearity.html.*

**2.3 Description of random effects**

You provide a nice description of your estimation approach, i.e., using a mixed effects ordinal model. However, for someone who is not used to these models, the random effects in the models are not completely clear.

Suggestion: I think that it would help the reader if you write out the specification of the different models in equation form. This would make it easier to understand the random effects in the models.

*Author response: Each result section that discusses mixed effects models describes the random effects included in the model both in the text and the tables that present the parameter estimates. We appreciate the reviewers request to make the model specification more accessible for readers less familiar with mixed effects models, but we doubt that writing out the specification of the different model in equation form would address the issue. Instead, we believe that more explicitly directing the interested reader to the R code of the analysis to see how the models are specified might be a more*

*approachable way to present the details of our model definitions (see added sentences with reference to our R code on p. 13 and 17). In addition, please note that we cite Harrison et al. (2018) in the data analysis section, which provides a very accessible explanation of the mixed effects models.*

**2.4 Experimental design**

As I understand it, you have two types of paired statements: 1) statements without extra explanation, with more or less jargon, 2) statements without jargon, with and without extra explanation. Together, you have four types of statements. In the regression in table 3, you use "more jargon" as the reference level. I have no problem interpreting the effect of reducing jargon, but I do struggle with interpreting the effect of "no added explanation" and "added explanation". How should I interpret these (insignificant) effects? If I have understood things correctly, then the "no added explanation" statements and "more jargon" statements represent distinct categories because the "no added explanation" statements do not contain jargon. However, the "no added explanation" is also different from "less jargon" because the original "no added explanation" statements did not include jargon. Have I understood this correctly? But then, how can you tell what it is that you are testing for here? It is quite possible that the problem is my lack of understanding. However, I would like to see a more elaborate description on how to interpret these results.

Suggestion: I think that it would be substantially easier to interpret the results if you split the sample between the two treatment types (jargon and explanation), i.e., that you treat your data as emanating from two different experiments.

*Author response: As pointed out by the reviewer, our experimental design included paired statements that either differed in the amount of jargon or explanation without any overlap. Our reason for originally combining the two treatments into a single variable was the desire to estimate only a single model since we expect the effect of the other predictor variables to be the same regardless of the treatment. Because of the larger sample size, the single model also has more statistical power to identify the effects of the other predictor variables.*

*However, the reviewer comment clearly highlights that this approach has resulted in confusing parameter estimates for the main and interaction effect associated with the statement treatment. We agree with the reviewer that estimating two separate models (one for jargon and one for explanation) will produce parameter estimates that are much cleaner and easier to interpret.*

*In response to this comment, we changed our analysis approach, which resulted in a substantial rewrite of the data analysis section (Section 2.3) and the entire result section (Section 3). While the nature of the results remains the same, the text has been completely rewritten.*

**2.5 Use of attention to TTA as a continuous variable**

Have I understood it correctly that you treat attention to the TTA as a continuous variable in the regression in table 3, while you treat it as an ordinal variable in table 2? What is your motivation for treating the variable in different ways in different models?

*Author response: While attention to TTA is the response variable in the model that is presented in Table 2, it is a predictor variable in all the subsequent models. In all models, we examined the parameter estimates of ordinal predictor variables and replaced them with numeric variables if the pattern indicated a linear relationship and the switch did not seem to negatively affect the interpretation. This*

*results in more parsimonious models. We now explain this aspect of our model estimation approach in the data analysis section (Section 2.3; p. 8).*

**2.6 Description of effects plots**

I really like that you illustrate the marginal effects in graphs. I also appreciate that you specify for which values you have estimated the marginal probabilities, and that you highlight that the reader should focus on significance levels and not predicted values. However, since the specifications used to estimate the marginal predictions are only given in footnotes, and since you talk about a broader group in the main text (e.g., people with introductory avalanche education), some readers may get the impression that your predictions are more general than they are.

Suggestion: Write out the specification in the text or in the figure caption. Since there are some variations in the values chosen to estimate the marginal effects, I also think that it would be beneficial if you include a brief description of why you have chosen these values. Don't get me wrong, I find the chosen values reasonable as they, in some sense, represent a representative participant.

*Author response: To be more transparent about the parameter values use for estimating the marginal probabilities, we now a) explicitly describe the parameter values in the text (e.g., section paragraph in Section 3.2 on p. 11), b) we added them in the captions of each figure (Figures 2, 3, 4, 5, 6, and 7), and c) we highlighted the levels by bolding them in the tables that present the parameter estimates (Tables B1, B2, B3, and B4).*

**2.7 Focus of discussion section**

The main aim of the study is to test if it is possible to improve the understanding and usefulness of the TTA. However, the manuscript is structured in a way that draws the attention to the effect of avalanche bulletin user type on the outcome variables. Given both the stated purpose of the paper, and the issues with this variable (the low number of type "A" users), I think that the structure is a bit unfortunate.

Suggestion: Restructure the text so that each result section starts with a discussion of the effects of the main explanatory variables (e.g., jargon and added explanation).

*Author response: We appreciate this comment, but it is important to note that our study aims to address three distinct research questions: a) who pays attention to the TTA statements, b) what factors determine the usefulness of these statements, and c) can modifications in the amount of jargon and explanations make the statements more useful. Hence, the description of the influence of participants' characteristics is important.*

*To ensure future readers will fully understand and appreciate the objective our study, the expanded the final paragraph of the introduction to be more explicit about our objective (p. 4), and we slightly modified the structure of the discussion section to better link it to the stated three objectives.*